# ITERKEY: Iterative Keyword Generation with LLMs for Enhanced Retrieval Augmented Generation

**Kazuki Hayashi†, Hidetaka Kamigaito†, Shinya Kouda‡, Taro Watanabe†**

†Nara Institute of Science and Technology ‡TDSE Inc.

{hayashi.kazuki.hl4, kamigaito.h, taro}@is.naist.jp

## Abstract

Retrieval Augmented Generation (RAG) has emerged as a way to complement the in-context knowledge of Large Language Models (LLMs) by integrating external documents. However, real-world applications demand not only accuracy but also interpretability. Dense retrieval methods provide high accuracy but lack interpretability, while sparse retrieval is transparent but often misses query intent due to keyword matching. Thus, balancing accuracy and interpretability remains a challenge. To address these issues, we introduce ITERKEY, an LLM-driven iterative keyword generation framework that enhances RAG via sparse retrieval. ITERKEY consists of three LLM-driven stages: generating keywords for retrieval, generating answers based on retrieved documents, and validating the answers. If validation fails, the process iteratively repeats with refined keywords. Across four QA tasks, experimental results show that ITERKEY achieves 5% to 20% accuracy improvements over BM25-based RAG and simple baselines. Its performance is comparable to dense retrieval based RAG and prior iterative query refinement methods using dense models. In summary, ITERKEY is a novel BM25-based iterative RAG framework that leverages LLMs to balance accuracy and interpretability.

## 1 Introduction

Large Language Models (LLMs) (OpenAI, 2023; Chiang et al., 2023; Dubey et al., 2024; Abdin et al., 2024) achieve strong performance in many natural language processing tasks but still face issues like hallucinations, outdated knowledge, and complex multi-hop reasoning (Kandpal et al., 2023; Zhang et al., 2023b; Gao et al., 2024). These challenges are pronounced in knowledge-intensive tasks (Lee et al., 2019; Zellers et al., 2018), where necessary information may not be fully stored or easily retrieved within model parameters.

Retrieval Augmented Generation (RAG) improves the accuracy and relevance of generated responses by integrating external knowledge, making it particularly effective for open domain question answering (Lewis et al., 2020; Izacard & Grave, 2021; Shuster et al., 2021; Ram et al., 2023; Izacard et al., 2024). Although improvements to individual RAG components such as query expansion, document reranking, and retrieval tuning have been extensively studied (Gao et al., 2024; Fan et al., 2024), real-world applications demand both interpretability and accuracy from the retrieval component. Dense retrieval methods are highly accurate but suffer from limited interpretability, while sparse retrieval methods are more transparent but often struggle to capture the underlying intent of user queries (Kang et al., 2025; Cheng et al., 2024; Ayoub et al., 2024; Llordes et al., 2023).

To address this, we propose ITERKEY: **Iter**ative **Key**word Generation with LLMs, a BM25-based method that improves RAG through iterative keyword refinement and self evaluation. ITERKEY operates in three stages: keyword generation, answer generation, and answer validation. In each cycle, an LLM extracts keywords, generates an answer, and checks correctness. If the validation fails, it updates the keywords and repeats the process until a satisfactory answer is obtained. By leveraging sparse BM25 retrieval, the contribution of

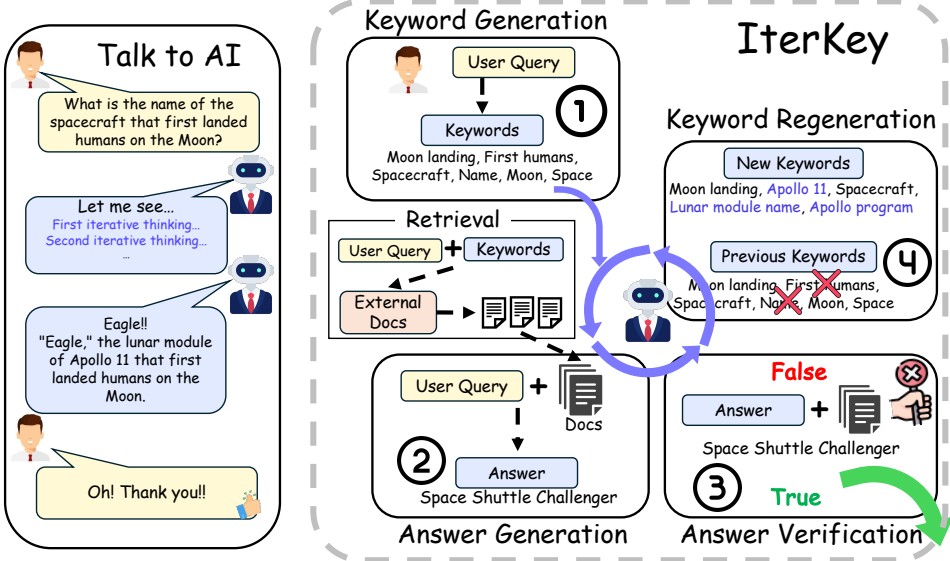

Figure 1: IterKey iteratively generates keywords, produces answers, and validates them using an LLM until the correct response is reached, thereby improving RAG accuracy.

each keyword to the score is explicit, enabling the tracking of generated keywords, retrieved documents, and validation results, which in turn improves accuracy and traceability.

Across four open domain QA datasets, ITERKEY improved retrieval performance by up to 20% over BM25 baselines and achieved 5% to 20% accuracy gains over vanilla methods and BM25-based RAG, with performance comparable to Dense model based RAG and iterative query refinement methods. These results indicate that LLMs can infer and expand keywords effectively, highlighting the practical potential of LLM-driven RAG refinement that balances performance and interpretability in supporting human search efforts.

## 2 ITERKEY: Iterative Keyword Generation with LLMs

Sparse retrieval algorithms, such as BM25 (Robertson & Zaragoza, 2009), efficiently handle large datasets by ranking documents based on keyword frequency and uniqueness. They are interpretable, fast, and training-free, making them suitable for scalable, low-resource settings. Their transparency lets users see which keywords influenced retrieval. However, they often miss nuanced or implicit query intent, limiting effectiveness compared to dense methods (Llordes et al., 2023; Izacard et al., 2022; Chen et al., 2023; Wang et al., 2022b). To address this limitation, we propose IterKey, a method that extends query expansion research (Wang et al., 2023a;b; Gao et al., 2023) by leveraging LLMs to iteratively refine the retrieval process in RAG. Specifically, IterKey uses the self-validation capabilities of LLMs (Wang et al., 2024) to improve sparse retrieval through iterative keyword generation, document retrieval, answer generation, and validation. By using BM25 for retrieval, all keywords that contribute to the search remain explicit, allowing the generated keywords, retrieved documents, and validation results to be inspected directly. This design aims to preserve the interpretability of sparse retrieval while improving accuracy. Table 1 summarizes the prompts used in each step of ITERKEY. We further explain the motivation behind each prompt design, with concrete examples provided in Figure 1 to illustrate the full process.

**Step 1: Keyword Generation**   Given a user query $q$, the LLM generates a set of critical keywords $\mathcal{K}^0$ essential for retrieving relevant documents. By leveraging BM25 retrieval and the query's intent, an LLM generates critical keywords from the initial query for retrieval. This step follows recent work on LLM-based query expansion (Xia et al., 2025; Lei et al., 2024) and enables the LLM to capture semantic and contextual keywords that sparse retrieval

| Step | Prompt |
|---|---|
| **Step 1: Initial Keyword Generation** | **System:** You are an assistant that generates keywords for information retrieval.
**User:** Generate a list of important keywords related to the **Query: { q }** .
Focus on keywords that are relevant and likely to appear in documents for BM25 search in the RAG framework.
Output the keywords as: ["keyword1", "keyword2", "keyword3", ...].
Separate each keyword with a comma and do not include any additional text. |
| **Step 2: Answer Generation using RAG** | **System:** You are an assistant that generates answers based on retrieved documents.
**User:** Here is a question that you need to answer: **Query: { q }**
Below are some documents that may contain information relevant to the question. Consider the information in these documents while combining it with your own knowledge to answer the question accurately.
Documents: { D }
Provide a clear and concise answer. Do not include any additional text. |
| **Step 3: Answer Validation** | **System:** You are an assistant that validates whether the provided answer is correct.
**User:** Is the following answer correct?
**Query: { q }**
Answer: { a }
Previous Retrieval Documents: { Docs }
Respond 'True' or 'False'. Do not provide any additional explanation or text. |
| **Step 4: Keyword Regeneration** | **System:** You refine keywords to improve document retrieval for BM25 search in the RAG framework.
**User:** Refine the keyword selection process to improve the accuracy of retrieving documents with the correct answer.
**Query: { q }**
Previous Keywords: { K }
Provide the refined list of keywords in this format: ["keyword1", "keyword2", ...].
Separate each keyword with a comma and do not include any additional text. |

Table 1: IterKey prompts for iterative keyword refinement and answer validation.

often misses, resulting in more effective document retrieval. In our example, for the query *"What is the name of the spacecraft that first landed humans on the Moon?"*, LLM generates important keywords like *Moon landing, Spacecraft, First humans*.

**Step 2: Answer Generation** The original query $q$ is expanded with the generated keywords $\mathcal{K}^i$ to form an enhanced query $q + \mathcal{K}^i$. This expanded query is used to retrieve a set of top-$k$ documents $\mathcal{D}^i = \{d_1, d_2, \ldots, d_k\}$ from an external corpus using a BM25-based retriever. Using the retrieved documents, the LLM generates an answer $a^i$ to the original query $q$. For example, the LLM generates the answer *Space Shuttle Challenger* using the expanded query.

**Step 3: Answer Validation** Recent studies (Asai et al., 2024; Yao et al., 2023) suggest that allowing an LLM to critique its own answer after generation can improve answer validity. Building on this insight, we let the LLM act as a judge to decide whether the generated answer is supported by the retrieved documents, using only the logical consistency between answer and evidence as the criterion (Zheng et al., 2023). The LLM is required to choose between 'True' or 'False' using force decoding, where the output is constrained to these two options. The generation probabilities for both are evaluated, and the option with the higher probability is selected as the answer. If 'True' is selected, the answer is considered correct, the process concludes, and $a^i$ is returned as the final answer. If 'False' is selected, the process restarts to refine the retrieval and generation steps. After reviewing the documents, the LLM finds *Space Shuttle Challenger* incorrect and returns 'False' in our example.

**Step 4: Keyword Regeneration** This step follows recent work on iterative relevance feedback with LLMs (Mackie et al., 2023; Liu et al., 2024). If the LLM responds with 'False' in the validation step, we regenerate a new set of keywords $\mathcal{K}^{i+1}$ to improve retrieval. The regeneration process uses the original query and the previous keywords as cues, allowing the LLM to refine its understanding of the query intent and progressively retrieve more relevant documents. The new keywords $\mathcal{K}^{i+1}$ are then used to form a new expanded query $q + \mathcal{K}^{i+1}$, and Steps 2 through 4 are repeated. This iterative process continues until the

validation step returns 'True' or a predefined maximum number of iterations $N$ is reached. For example, after receiving 'False', the LLM generates new keywords like *Apollo 11, Lunar module name*, and the correct answer *Eagle* is finally retrieved in Figure 1.

## 3 Experiments

### 3.1 Settings

**Datasets and Evaluation Methods** We evaluated ITERKEY on four open domain QA datasets: Natural Questions (NQ) (Kwiatkowski et al., 2019), EntityQA (Li et al., 2019), We­bQA (Chang et al., 2022), and HotpotQA (Yang et al., 2018). Following prior work (Trivedi et al., 2023; Jiang et al., 2023b; Feng et al., 2024), we randomly sampled 500 entries per dataset due to computational cost and conducted the evaluation under zero-shot settings. We evaluate generated answers using the Exact Match (EM) metric (Rajpurkar et al., 2016), which considers an answer correct if it matches any reference after normalization (lower­casing, removing articles and punctuation, and whitespace consolidation). To assess the impact of query expansion, we compute recall as the percentage of retrieved documents that contain at least one reference answer. We used the December 2018 Wikipedia dump as the retrieval corpus for all datasets (Izacard et al., 2024) following (Feng et al., 2024).

**Retrieval Models** We utilized BM25 (Robertson & Zaragoza, 2009), implemented using BM25S (Lù, 2024)[1], as the base retriever. Additionally, we adopted three Dense Models: Contriever (Izacard et al., 2022), BGE (Chen et al., 2023), and E5 (Wang et al., 2022b), under identical conditions to compare their retrieval performance against sparse models.

**LLMs** To examine the practical utility of ITERKEY across different model settings, we evaluate it using four LLMs with varying capabilities: Llama-3.1 8B and 70B (Dubey et al., 2024), Gemma-2 (Team et al., 2024), and Phi-3.5-mini (Abdin et al., 2024). This analysis examines how generation and validation abilities impact each component, offering insights for practical model selection. All models can produce the structured outputs required by each ITERKEY step. See Appendices B and C for details.

**Comparative Baseline** As a representative iterative baseline, we adopt ITRG (Iterative Retrieval-Generation Synergy) (Feng et al., 2024), the strongest publicly available method in this line of work. ITRG repeatedly concatenates LLM-generated answer passages with the original query to refine retrieval quality. In contrast, IterKey generates explicit keyword lists and introduces an answer validation step that halts iteration once the answer is deemed correct. ITRG integrates established single-pass query expansion techniques (Wang et al., 2023b;a; Gao et al., 2023), positioning it as an appropriate, advanced comparator.

### 3.2 Results

**IterKey vs. Baseline & RAG (BM25)** Table 2 shows that ITERKEY consistently improves accuracy over the Baseline across all models, achieving gains of 10% to 25%. Particularly, Llama-3.1 models of 8B and 70B parameters show notable improvements with ITERKEY, with the 8B model reaching accuracy levels similar to the 70B model. This demonstrates ITERKEY's effectiveness in enhancing the performance of smaller models, allowing them to achieve accuracy levels closer to larger ones. Further comparison with BM25-based RAG reveals a 5% to 10% accuracy increase on Llama-3.1 when using ITERKEY. However, ITERKEY performs comparably to RAG (BM25) on Phi-3.5-mini, while falling behind RAG (BM25) on Gemma-2, with no notable improvements on the other tasks. These findings highlight the effectiveness of ITERKEY, although the benefits vary across models.

**IterKey vs. RAG (E5) & ITRG** As shown in Table 2, ITERKEY with BM25-based retrieval outperforms E5 based RAG on Llama-3.1 models. Notably, the Llama-3.1 70B model achieves

---

[1] https://github.com/xhluca/bm25s

| Method | Model | Size (B) | Entity QA | HotpotQA | Natural QA | WebQA |
|---|---|---|---|---|---|---|
| **Vanilla** | Llama-3.1 | 8B | 33.6 | 31.2 | 40.6 | 53.4 |
| | Llama-3.1 | 70B | 45.2 | 41.4 | 46.0 | 54.0 |
| | Gemma-2 | 9B | 10.6 | 11.6 | 9.2 | 20.8 |
| | Phi-3.5-mini | 3.8B | 24.6 | 25.4 | 25.8 | 44.0 |
| **RAG (BM25)** | Llama-3.1 | 8B | 54.0 | 47.0 | 44.8 | 51.4 |
| | Llama-3.1 | 70B | 54.6 | 46.2 | 43.4 | 47.4 |
| | Gemma-2 | 9B | 47.9 | 39.6 | 33.2 | 41.6 |
| | Phi-3.5-mini | 3.8B | 48.2 | 42.2 | 32.6 | 40.2 |
| **RAG (E5)** | Llama-3.1 | 8B | 52.9 | 47.7 | 49.6 | 48.2 |
| | Llama-3.1 | 70B | 57.0 | 51.0 | 49.4 | 48.8 |
| | Gemma-2 | 9B | 52.2 | 40.8 | 41.5 | 41.8 |
| | Phi-3.5-mini | 3.8B | 50.2 | 44.6 | 37.0 | 41.4 |
| **ITRG Refresh (E5)** | Llama-3.1 | 8B | 60.6 | 53.4 | 53.6 | 56.2 |
| | Llama-3.1 | 70B | 60.7 | 52.9 | 53.3 | 51.6 |
| | Gemma-2 | 9B | 54.2 | 47.6 | 47.4 | 48.5 |
| | Phi-3.5-mini | 3.8B | 54.3 | 47.1 | 36.2 | 44.6 |
| **IterKey (BM25)** | Llama-3.1 | 8B | 62.0 | 49.7 | 50.2 | 53.3 |
| | Llama-3.1 | 70B | **63.5** | **54.9** | **53.8** | **57.7** |
| | Gemma-2 | 9B | 36.1 | 26.1 | 35.9 | 36.1 |
| | Phi-3.5-mini | 3.8B | 47.9 | 42.7 | 36.8 | 41.2 |

Table 2: 'Vanilla' uses no retrieval. 'RAG (BM25)' and 'RAG (E5)' perform a single retrieval step on the original query using BM25 and E5, respectively. Other dense retriever results are in Table 18. 'ITRG Refresh (E5)' is a prior iterative refinement method, reproduced here with E5 (its best dense retriever) and five query expansions. Two approaches, Refine and Refresh, exist; see (Feng et al., 2024) for details. Refine results are in Table 19, while Refresh is in the main text. Our 'IterKey (BM25)' iteratively generates and refines keywords for up to five retrieval iterations to optimize answer quality. In the table, underlined values mark the best performance for each model, and **bold values** the highest accuracy overall.

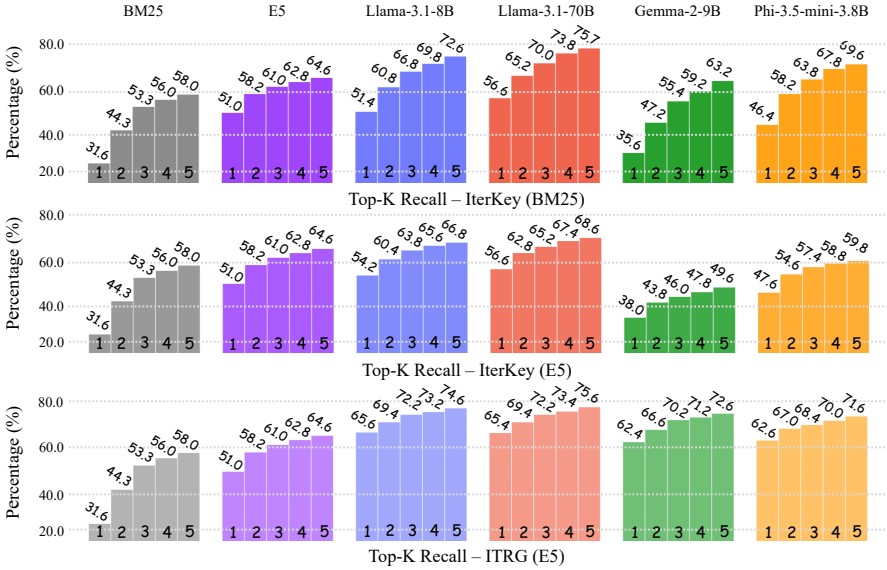

Figure 2: The figure shows the Top-K retrieval recall rates for each model in the Entity QA task, representing the proportion of retrieved documents containing the correct answer. The baseline methods, 'BM25' and 'E5,' use a single retrieval step based on the original query. Our proposed method, 'IterKey (BM25),' is compared with 'IterKey (E5)' for dense retrieval and the prior work 'ITRG (E5).' Recall rates are averaged over five iterations.

the highest accuracy across all tasks, outperforming ITRG, an iterative refinement method that employs a dense retriever. Moreover, the Llama-3.1 8B model attains accuracy comparable to ITRG, demonstrating that ITERKEY can effectively compete with dense retrieval based iterative strategies even at smaller scales. However, ITERKEY's effectiveness does not

extend uniformly across all models. While ITRG consistently improves performance for Gemma-2 and Phi-3.5-mini, ITERKEY does not achieve comparable gains in these models. Additionally, compared to E5 based RAG, its performance remains lower across most tasks.

## 4 Analysis

### 4.1 Recall and QA Performance in IterKey

Figure 2 shows recall rates (%) on the Entity QA task for three methods and four models, averaged over five iterations. For comparison, BM25 and E5 (leftmost) show a single retrieval step using the original query. Recall is the percentage of retrieved documents containing the correct answer. IterKey (BM25) improves recall by approximately 20% over standard BM25. For the Llama-3.1 models, the top-5 recall is about 10% higher than that of E5, indicating that LLMs effectively generate and expand keywords to enhance retrieval. Although Gemma and Phi outperform BM25, their recall remains below E5, implying keyword generation quality depends heavily on the model. Despite achieving approximately 60% top-3 recall, Gemma and Phi exhibit QA accuracy that is 20–30 percentage points lower (see Table 2), suggesting weaknesses in answer generation or validation significantly reduce performance.

Comparing IterKey (BM25) and IterKey (E5) reveals similar Top-1 performance; however, IterKey (BM25) achieves 10% higher Top-5 precision and higher overall QA accuracy, see Table 6. This likely reflects that plain keyword lists align well with BM25, while E5, trained for natural-language queries, handles keyword strings less effectively. These results show that our iterative keyword refinement is especially effective for sparse retrieval models.

To further analyze how accuracy evolves over iterations, Table 5 reports per-iteration results. Most performance gains occur in the first one or two iterations, with accuracy improving by up to 7%. After the third iteration, improvements taper off, indicating diminishing returns. This demonstrates that IterKey balances efficiency and effectiveness by capturing most of the benefits early while retaining the flexibility to continue when necessary.

| Model | Step 2 | Step 3 | Step 4 | Step 5 | Total | Mean |
|---|---|---|---|---|---|---|
| Llama-3.1-8B | 10.5 | 9.57 | 8.59 | 8.20 | 36.9 | 9.21 |
| Llama-3.1-70B | 9.34 | 3.65 | 3.78 | 3.26 | 20.0 | 5.01 |
| Gemma-2 | 14.1 | 8.56 | 5.70 | 4.59 | 33.0 | 8.25 |
| Phi-3.5-mini | 18.1 | 7.12 | 6.00 | 5.07 | 36.3 | 9.07 |

| Model | Step 2 | Step 3 | Step 4 | Step 5 | Total | Mean |
|---|---|---|---|---|---|---|
| Llama-3.1-8B | 2.12 | 1.65 | 1.53 | 1.32 | 6.62 | 1.66 |
| Llama-3.1-70B | 1.93 | 1.44 | 1.15 | 1.25 | 5.77 | 1.44 |
| Gemma-2 | 2.22 | 1.61 | 1.27 | 1.12 | 6.22 | 1.55 |
| Phi-3.5-mini | 2.03 | 0.69 | 1.04 | 0.95 | 4.71 | 1.18 |

Table 3: Average number of new keywords generated per query at each iteration.

Table 4: Average number of new documents retrieved per query at each iteration.

### 4.2 Keyword and Document Expansion by Iteration

To clarify the role of each iteration in our method, we show the average number of new keywords and documents added at each step. Table 3 presents the average number of newly generated keywords per query at each iteration from Step 2 to Step 5. Table 4 shows the corresponding statistics for newly retrieved documents. These results show that each iteration continues to introduce new information. On average, each step generates between 3 and 18 additional keywords per query, resulting in a total of 20 to 37 new keywords after four steps. Similarly, each iteration retrieves approximately 0.7 to 2.2 new documents per query, with a total of 4.7 to 6.6 documents accumulated through the full process. These findings indicate that later iterations are effective in generating novel keywords and retrieving previously unseen documents. The iterative process enables the method to maintain both efficiency and coverage, as it allows early termination for easier queries while providing additional refinement for more challenging cases.

### 4.3 Keyword Generation Step

Based on these results, we conducted additional experiments to further analyze the impact of keyword quality on ITERKEY's performance. Specifically, we evaluated the effects of

| Model | Iter 1 | Iter 2 | | Iter 3 | | Iter 4 | | Iter 5 | |
|---|---|---|---|---|---|---|---|---|---|
| Llama-3.1 8B | 62.2 | 65.4 | +3.2 | 69.6 | +7.4 | 70.5 | +8.3 | 72.6 | +10.4 |
| Llama-3.1 70B | 65.7 | 70.2 | +4.5 | 73.5 | +7.8 | 74.4 | +8.7 | 75.7 | +10.0 |
| Gemma-2 9B | 53.2 | 57.6 | +4.4 | 60.0 | +6.8 | 61.2 | +8.0 | 63.2 | +10.0 |
| Phi-3.5 mini | 57.5 | 64.5 | +7.0 | 67.9 | +10.4 | 68.8 | +11.3 | 69.6 | +12.1 |

Table 5: Accuracy (%) on the Entity QA task for each iteration. Values in blue boxes show absolute gains from Iter 1.

| Method | Model | Size (B) | Entity QA | | HotpotQA | | Natural QA | | WebQA | |
|---|---|---|---|---|---|---|---|---|---|---|
| **IterKey (BM25)** | Llama-3.1 | 8B | | 62.0 | | 49.7 | | 50.2 | | 53.3 |
| | Llama-3.1 | 70B | | 63.5 | | 54.9 | | 53.8 | | 57.7 |
| | Gemma-2 | 9B | | 36.1 | | 26.1 | | 35.9 | | 36.1 |
| | Phi-3.5-mini | 3.8B | | 47.9 | | 42.7 | | 36.8 | | 41.2 |
| **IterKey (E5)** | Llama-3.1 | 8B | -1.8 | 60.2 | +4.1 | 53.8 | +1.2 | 51.4 | -2.7 | 50.6 |
| | Llama-3.1 | 70B | -1.3 | 62.2 | -2.1 | 52.8 | -0.6 | 53.2 | -5.7 | 52.0 |
| | Gemma-2 | 9B | -1.6 | 34.5 | -0.9 | 25.2 | -0.7 | 35.2 | -3.7 | 32.4 |
| | Phi-3.5-mini | 3.8B | -0.5 | 47.4 | +1.8 | 44.5 | -2.4 | 34.4 | -1.0 | 40.2 |

Table 6: Performance of ITERKEY with 'BM25' and the Dense Model ('E5'). This table shows the QA performance when replacing 'BM25' with 'E5' under consistent settings.

using high quality keywords generated by Llama 3.1 70B, which achieved the highest recall, and low quality keywords from Gemma 2, which had the lowest recall. As shown in Table 7, using high-quality keywords consistently improved accuracy across all models, notably resulting in an about 5% gain for Gemma 2. Conversely, employing low-quality keywords degraded performance for all models and tasks, with Llama 3.1 (8B) experiencing a particularly significant accuracy drop of 4.5% on EntityQA. These results clearly demonstrate that keyword quality plays a crucial role in ITERKEY's performance.

Notably, we observed that even when provided with high-quality keywords, Gemma-2's accuracy still lagged considerably behind Llama-3.1. Given that ITERKEY iteratively refines retrieval and validation, this suggests that Gemma-2's primary performance bottleneck lies in its validation capability rather than in keyword generation.

## 4.4 Answer Validation Step

We conduct additional experiments to examine validation model quality affects IterKey performance. We compare the high quality validation model Llama 3.1 70B, which achieves the highest baseline accuracy, with the low quality validation model Gemma, which shows the lowest baseline accuracy. As shown in Table 5, using a high quality validation model consistently improves accuracy across all base models. Gemma improves by more than five points on average over four tasks. In contrast, Gemma lowers accuracy for all models, with LLaMA-based models dropping by over ten points, showing validation quality is critical for IterKey and that Gemma's limited validation capability severely constrains its performance.

We also examined how IterKey's Answer Validation step performs over repeated iterations. Specifically, we iterated up to five times, including after a 'True' judgment, and compared accuracy across three settings to assess each model's validation performance. As shown in Table 9, comparing VerifiedAll' with Base' reveals substantial discrepancies in validation accuracy across all models, with Gemma-2 showing the largest gaps (16.9% in HotpotQA and 15.2% in Entity QA). These results indicate that validation errors are more prominent in weaker models, highlighting the challenge of ensuring reliable answer verification.

The miss True rate ('Base' vs. 'VerifiedTrue') measures the tendency to accept wrong answers as correct, while the miss False rate ('VerifiedTrue' vs. 'VerifiedAll') quantifies failure to recognize correct answers under stricter validation. For Llama-3.1-8B, Llama-3.1-70B, and Phi-3.5-mini, 'miss True' is consistently higher than 'miss False,' indicating a tendency to

| Method | Model | Size (B) | Entity QA | HotpotQA | Natural QA | WebQA |
|---|---|---|---|---|---|---|
| **IterKey** | Llama-3.1 | 8B | 62.0 | 49.7 | 50.2 | 53.3 |
| | Llama-3.1 | 70B | 63.5 | 54.9 | 53.8 | 57.7 |
| | Gemma-2 | 9B | 36.1 | 26.1 | 35.9 | 36.1 |
| | Phi-3.5-mini | 3.8B | 47.9 | 42.7 | 36.8 | 41.2 |
| **IterKey w/ HQ Keywords** | Llama-3.1 | 8B | +1.4 63.4 | +3.1 52.8 | +1.4 51.6 | +0.7 54.0 |
| | Llama-3.1 | 70B | 63.5 | 54.9 | 53.8 | 57.7 |
| | Gemma-2 | 9B | +3.9 40.0 | +4.4 30.5 | +1.6 37.5 | +3.9 40.0 |
| | Phi-3.5-mini | 3.8B | +1.9 49.8 | +1.5 44.2 | +1.2 38.0 | +2.2 43.4 |
| **IterKey w/ LQ Keywords** | Llama-3.1 | 8B | -6.0 56.0 | -0.5 49.2 | 50.2 | -1.4 51.9 |
| | Llama-3.1 | 70B | -6.4 57.1 | -2.6 52.3 | -3.3 50.5 | -5.1 52.6 |
| | Gemma-2 | 9B | 36.1 | 26.1 | 35.9 | 36.1 |
| | Phi-3.5-mini | 3.8B | -2.8 45.1 | -2.2 40.5 | -2.4 34.4 | +0.2 41.4 |

Table 7: Ablation study on keyword quality in ITERKEY. 'High-quality (HQ) keywords' are generated by Llama-3.1-70B, the model with the highest retrieval recall, and reused by smaller models with up to 10× fewer parameters, consistently boosting performance. 'Low-quality (LQ) keywords,' generated by Gemma-2-9B, degrade performance across models.

| Method | Model | Size (B) | Entity QA | HotpotQA | Natural QA | WebQA |
|---|---|---|---|---|---|---|
| **IterKey** | Llama-3.1 | 8B | 62.0 | 49.7 | 50.2 | 53.3 |
| | Llama-3.1 | 70B | 63.5 | 54.9 | 53.8 | 57.7 |
| | Gemma-2 | 9B | 36.1 | 26.1 | 35.9 | 36.1 |
| | Phi-3.5-mini | 3.8B | 47.9 | 42.7 | 36.8 | 41.2 |
| **IterKey w/ HQ Validation Model** | Llama-3.1 | 8B | +0.5 62.5 | +2.0 51.7 | +2.0 52.2 | +1.2 54.5 |
| | Llama-3.1 | 70B | 63.5 | 54.9 | 53.8 | 57.7 |
| | Gemma-2 | 9B | +7.4 43.5 | +8.0 34.1 | +4.6 40.5 | +3.3 39.4 |
| | Phi-3.5-mini | 3.8B | +2.9 50.8 | +1.9 44.6 | +1.0 37.8 | +2.2 43.4 |
| **IterKey w/ LQ Validation Model** | Llama-3.1 | 8B | -10.4 51.6 | -8.0 41.7 | -5.7 44.5 | -7.1 46.2 |
| | Llama-3.1 | 70B | -10.7 52.8 | -11.9 43.0 | -11.4 42.4 | -5.9 51.8 |
| | Gemma-2 | 9B | 36.1 | 26.1 | 35.9 | 36.1 |
| | Phi-3.5-mini | 3.8B | -2.6 45.3 | -2.6 40.1 | -4.0 32.8 | -3.0 38.2 |

Table 8: Performance comparison of the IterKey validation step. The 'High Quality' uses Llama-3.1-70B for the validation step, whereas the 'Low Quality' employs Gemma, the model with the lowest baseline accuracy. Only the validation step is modified, and the performance changes across four tasks (improvements in blue, degradations in red) are shown.

accept incorrect answers and limiting accuracy gains. In contrast, Gemma-2 shows a higher 'miss False' rate, reflecting difficulty identifying correct answers under stricter validation.

Overall, these results show that validation errors, particularly 'miss True,' significantly constrain accuracy improvements. Thus, a model's validation capability is essential when applying ITERKEY. Moreover, adherence to output format does not necessarily imply strong instruction comprehension. Even the models that follow formatting constraints may fail to apply validation rules accurately, as seen in Gemma-2's validation errors. This aligns with prior studies (Kung & Peng, 2023; Zhou et al., 2023b; Liang et al., 2024) emphasizing the distinction between format adherence and instruction comprehension in LLMs.

## 4.5 Keyword Regeneration Prompt

We compared IterKey (BM25), which reuses previous keywords as a proxy for documents in subsequent iterations, with a document-by-document approach that regenerates and concatenates keywords for each retrieved document for the next retrieval. Since we use three documents at retrieval time, in a single iteration, the model ends up generating keywords two additional times. The prompt we used is shown in Table 10.

As shown in Table 11, the approach that generates keywords based on each retrieved document achieves higher recall for Gemma-2. However, for Llama-3.1 (8B, 70B) and Phi-3.5-mini, our original IterKey (BM25) approach still produces higher accuracy. Moreover, from a cost performance perspective, using previously generated keywords as a proxy for

| Setting | Model | Size (B) | Entity QA | | HotpotQA | | Natural QA | | WebQA | |
|---------|-------|----------|-----------|---|----------|---|------------|---|-------|---|
| **Base** | Llama-3.1 | 8B | | 62.0 | | 49.7 | | 50.2 | | 53.3 |
| | Llama-3.1 | 70B | | 63.5 | | 54.9 | | 53.8 | | 57.7 |
| | Gemma-2 | 9B | | 36.1 | | 26.1 | | 35.9 | | 36.1 |
| | Phi-3.5-mini | 3.8B | | 47.9 | | 42.7 | | 36.8 | | 41.2 |
| **VerifiedTrue** | Llama-3.1 | 8B | +1.9 | 63.9 | +5.3 | 55.0 | +4.3 | 54.5 | +4.5 | 57.8 |
| | Llama-3.1 | 70B | +2.3 | 65.8 | +2.1 | 57.0 | +4.0 | 57.8 | +4.7 | 62.4 |
| | Gemma-2 | 9B | +4.9 | 41.0 | +8.1 | 34.2 | +3.5 | 39.4 | +4.4 | 40.5 |
| | Phi-3.5-mini | 3.8B | +4.6 | 52.5 | +4.4 | 47.1 | +3.7 | 40.5 | +5.6 | 46.8 |
| **VerifiedAll** | Llama-3.1 | 8B | +4.6 | 66.6 | +8.9 | 58.6 | +10.0 | 60.2 | +7.5 | 60.8 |
| | Llama-3.1 | 70B | +4.3 | 67.8 | +6.6 | 61.5 | +10.2 | 64.0 | +6.5 | 64.2 |
| | Gemma-2 | 9B | +14.3 | 50.4 | +16.4 | 42.5 | +7.9 | 43.8 | +11.5 | 47.6 |
| | Phi-3.5-mini | 3.8B | +7.7 | 55.6 | +7.5 | 50.2 | +8.6 | 45.4 | +11.2 | 52.4 |

Table 9: Accuracy results for different models and settings across four QA tasks. **Settings**: 'Base' uses the first "True" in the Answer Verification step as the answer, then stops. 'VerifiedTrue' checks if any iteration judged "True" contains the correct answer. 'VerifiedAll' checks all iterations, both "True" and "False," for the correct answer.

| Step | Prompt |
|------|--------|
| **Step 4: Keyword Regeneration with Docs** | **System:** Refine the provided keywords to enhance document retrieval accuracy for BM25 search in the RAG framework.
**User:** Please refine the keyword selection process to improve the accuracy of retrieving documents containing the correct answer.
**Query:** { *q* }
Previous Keywords: { *K* }
Previous Retrieval Documents: { *Docs* }
Provide the refined list of keywords in this format: ["keyword1", "keyword2", ...].
Ensure each keyword is separated by a comma, and do not include any additional text. |

Table 10: Compared to the Step 4 prompt in Table 1, this prompt incrementally adds 'Previous Retrieval Documents' at a time and regenerates new keywords at each step.

documents remains more efficient. Hence, while the document-by-document approach is effective for Gemma-2, our method of using previously generated keywords as a proxy offers a more favorable balance of accuracy and cost in most cases.

## 4.6 Computational Cost

We evaluated how well ITERKEY balances accuracy and computational cost compared to other methods. Table 12 compares runtime for the Entity QA dataset using Llama-3.1-8B. For a fair comparison, both ITERKEY and ITRG used E5 for retrieval. Inference was performed on an NVIDIA RTX 6000 Ada GPU, while BM25 retrieval used an Intel Xeon Platinum 8160 CPU. As a result, IterKey achieved comparable accuracy to ITRG while using only approximately 80% of the latency per question, with an average latency of 3.43 seconds.

Table 13 shows the average iterations required by ITERKEY across all QA tasks and models. The count stays below 1.5, with each iteration involving three LLM calls: keyword generation, answer generation, and validation. Multiplying the 3.43 s base latency by the average 1.5 iterations yields $\approx$ 5.15 s per question. In summary, while iterative reasoning introduces a certain overhead (about 4 seconds slower than non-iterative baselines), the per-question latency remains well within acceptable bounds for real-time QA with GPU-based inference. Given the substantial improvements in accuracy and robustness, we consider this a reasonable trade-off in practical settings where reliability and interpretability are essential.

## 5 Related Work

**Retrieval Augmented Generation (RAG)** RAG enhances response accuracy and relevance by incorporating external knowledge, making it effective for open-domain QA (Lewis et al., 2020; Izacard & Grave, 2021; Shuster et al., 2021; Izacard et al., 2024). It also addresses the limitations of in-context learning, which struggles with complex questions and outdated information (Liu et al., 2022; Petroni et al., 2019; Borgeaud et al., 2022; Roberts et al., 2020).

| Method | Model | Size (B) | Top1 (%) | Top2 (%) | Top3 (%) | Top4 (%) | Top5 (%) |
|---|---|---|---|---|---|---|---|
| IterKey (BM25) | Llama-3.1 | 8B | 51.4 | 60.8 | 66.8 | 69.8 | 72.6 |
| | Llama-3.1 | 70B | 56.6 | 65.2 | 70.0 | 73.8 | 75.7 |
| | Gemma-2 | 9B | 35.6 | 47.2 | 55.4 | 59.2 | 63.2 |
| | Phi-3.5-mini | 3.8B | 46.4 | 58.2 | 63.8 | 67.8 | 69.6 |
| IterKey (BM25) document-by-document | Llama-3.1 | 8B | +3.6 55.0 | -0.8 60.0 | -3.4 63.4 | -4.6 65.2 | -4.6 68.0 |
| | Llama-3.1 | 70B | +2.4 59.0 | +0.2 65.4 | -1.8 68.2 | -3.4 70.4 | -2.9 72.8 |
| | Gemma-2 | 9B | +15.4 51.0 | +8.8 56.0 | +2.8 58.2 | +3.0 62.2 | +2.2 65.4 |
| | Phi-3.5-mini | 3.8B | +2.4 48.8 | -4.2 54.0 | -8.0 55.8 | -7.2 60.6 | -6.4 63.2 |

Table 11: Retrieval performance comparison between the original IterKey (BM25) and the document-by-document keyword regeneration approach.

| Step | RAG(BM25) | RAG(E5) | ITRG | IterKey |
|---|---|---|---|---|
| Query Expansion | – | – | – | 1.33 |
| Retrieval | 0.44 | 0.11 | 0.88 | 0.33 |
| Answer Generation | 0.64 | 0.63 | 3.39 | 1.05 |
| Answer Validation | – | – | – | 0.72 |
| Total | 1.08 | 0.73 | 4.26 | 3.43 |

| Model | Size (B) | Entity | Hotpot | Natural | Web |
|---|---|---|---|---|---|
| Llama-3.1 | 8B | 1.27 | 1.38 | 1.35 | 1.36 |
| Llama-3.1 | 70B | 1.15 | 1.20 | 1.23 | 1.10 |
| Gemma-2 | 9B | 1.33 | 1.36 | 1.34 | 1.37 |
| Phi-3.5-mini | 3.8B | 1.38 | 1.32 | 1.28 | 1.29 |

Table 12: Average latency (s) for each step in Entity QA with LLaMA-3.1-8B.

Table 13: Average iterations for ITERKEY.

Recent work has focused on improving components such as query expansion, retrieval tuning, summarization, and reranking (Gao et al., 2024; Fan et al., 2024). RAG's effectiveness depends on both its components and their interaction (Jiang et al., 2022; Zhang et al., 2023a), yet these processes often function independently, making coherence difficult (Shi et al., 2024; Guo et al., 2023). Even when relevant documents are retrieved, they may not be fully used in generating responses (Ram et al., 2023; Dai et al., 2023; Xu et al., 2024; Asai et al., 2024), and retrieval failures can lead to irrelevant outputs (Jiang et al., 2023b; Guo et al., 2023).

**Query and Document Expansion** Recent methods improve retrieval by generating pseudo documents from queries, benefiting both sparse and dense systems (Wang et al., 2023a). Generating hypothetical documents has also enhanced zero-shot dense retrieval without relevance labels (Gao et al., 2023). These approaches leverage LLMs to address short or ambiguous queries by adding relevant context (Zhang et al., 2023b; Asai et al., 2023).

**Iterative and Chain-of-Thought Retrieval** Iterative retrieval methods leverage intermediate reasoning to refine queries for complex, multi-step information needs, integrating generative and Chain-of-Thought (CoT) reasoning to dynamically update queries and responses (Shao et al., 2023; Kim et al., 2023; Feng et al., 2024; Sun et al., 2023; Jiang et al., 2023b; Arora et al., 2023; Wang et al., 2022a). CoT-guided retrieval further enhances document relevance, factual accuracy, and reduces hallucinations (Trivedi et al., 2023; Creswell et al., 2023; Zhou et al., 2023a; Yao et al., 2023; Wei et al., 2022; Kojima et al., 2024; Zheng et al., 2024). Both supervised and unsupervised iterative approaches excel in multi-hop QA and knowledge-intensive tasks, making them effective even in few-shot and zero-shot scenarios (Das et al., 2019; Xiong et al., 2021; Khattab et al., 2023).

# 6 Conclusion

We introduce ITERKEY, a novel BM25-based framework that refines Retrieval Augmented Generation (RAG) by dividing the process into three stages: keyword generation, answer generation, and answer validation, all performed iteratively by an LLM. Experimental results across four open domain QA datasets show that ITERKEY improves retrieval performance by 20% over standard BM25 baselines and achieves 5% to 20% accuracy gains over vanilla and BM25-based RAG. It also delivers performance comparable to Dense model-based RAG and prior iterative refinement methods. These results demonstrate that LLMs can infer and refine keywords effectively to improve retrieval and answer quality, while preserving the transparency of sparse retrieval. Our findings highlight the potential of LLMs to enhance both accuracy and interpretability in real-world RAG applications.

## Ethical Considerations

Language models are known to sometimes generate incorrect or potentially biased information. This issue is particularly significant when sensitive questions are posed to the model. While our retrieval-augmented approach is expected to mitigate this problem to some extent by grounding responses in external sources, it does not fully eliminate the risk of biased or offensive outputs. Therefore, careful consideration is required when deploying such systems in user facing applications to avoid unintended harm. All datasets and models used in this work are publicly available under permissible licenses. The Natural Questions dataset is provided under the CC BY-SA 3.0 license, and the WebQuestions dataset is also available under the CC BY-SA 3.0 license. The data related to EntityQA are distributed according to their respective licensing terms. Additionally, the use of these datasets is permitted for research purposes. Note that in this work, we used an AI assistant tool, ChatGPT, for coding support.

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

## A    Limitations

The performance of ITERKEY is highly dependent on the capabilities of the underlying large language model (LLM). For example, Gemma-2, a recent high-performance model, showed a notable drop in accuracy despite adhering to the output format and instructions. This suggests that even advanced models may struggle to effectively perform keyword generation or answer validation, limiting the range of LLMs suitable for ITERKEY. This aligns with prior studies (Kung & Peng, 2023; Zhou et al., 2023b; Liang et al., 2024) emphasizing the distinction between format adherence and instruction comprehension in LLMs.

ITERKEY introduces additional computational cost compared to non-iterative RAG methods. Each iteration involves keyword generation, retrieval, answer generation, and validation, resulting in higher runtime and resource usage, especially with large models or in large-scale applications. Additionally, since ITERKEY operates under a fixed iteration limit, it may fail to resolve highly ambiguous or complex queries within the allotted steps.

Finally, the answer validation step relies on textual pattern matching, which can misjudge semantically correct answers expressed in different formats such as abbreviations or paraphrases. As a result, true positives may be overlooked, leading to underestimation of model performance. This limitation particularly affects the accuracy of the ablation analysis, where validation quality is a central factor.

In this study, to ensure a fair comparison of performance across multiple models, all experiments were conducted on a single NVIDIA RTX 6000 Ada GPU, with half quantization utilized for model generation. However, due to resource constraints, the Llama-3.1 70B model was loaded and inferred in 4-bit mode. We use LLMs for three steps: keyword generation, answer generation, and answer validation. For each step, an appropriate Max Token Length was set, as shown in Table 14. The same settings were applied to each model for performance comparison purposes.

Table 15 provides details of the language models used in our experiments, including their parameter sizes and Hugging Face model identifiers. These models were selected based on their performance and availability for fair evaluation across different LLM architectures.

| STEP | Max Token Length |
|---|---|
| Keyword Generation | 50 |
| Answer Generation | 50 |
| Answer Validation | 30 |

Table 14: Max token length for each step.

| Model | Params | HuggingFace Name |
|---|---|---|
| Llama-3.1 | 8B | meta-llama/Llama-3.1-8B-Instruct |
| Llama-3.1 | 70B | meta-llama/Llama-3.1-70B-Instruct |
| Gemma-2 | 9B | google/Gemma-2-9b-it |
| Phi-3.5-mini | 3.8B | microsoft/Phi-3.5-mini-instruct |

Table 15: Details of the LMs for the experiments.

## B  Details of Experimental Setting

## C  Selection of Models for Iterkey

For the selection of models used in Iterkey, we conducted preliminary experiments with multiple candidate models. Based on the results, we selected models according to two main criteria (Section 2) required to implement our method without human intervention.

1. **Adherence to Specified Output Format:** The model must follow the designated generation format. In keyword generation and validation, it should produce outputs that conform to the specified format with minimal post-processing, generating lists without unnecessary extra text.

2. **Accurate Understanding of Instructions:** The model must accurately comprehend the intent of the instructions. It should generate outputs based on a correct understanding of the instructions. During validation and keyword regeneration, it should produce consistent outputs aligned with the task requirements.

Models such as Llama 2 (Touvron et al., 2023), Qwen 1.5 (Bai et al., 2023), Vicuna (Chiang et al., 2023), and Mistral (Jiang et al., 2023a) were not adopted for ITERKEY as they either produced erroneous outputs or failed to adhere to the specified format, making them unsuitable for use in this context. Additionally, all selected models received instruction tuning, as models without instruction tuning were unable to generate controlled outputs effectively.

## D  Effect of Chain of Thought Reasoning

To examine whether explicit Chain of Thought (CoT) reasoning can further boost ITERKEY, we replaced the original single sentence prompts in Answer Validation (Step 3) and Keyword Regeneration (Step 4) with step by step versions. Specifically, in Step 3 the model first writes out its reasoning, citing the retrieved documents, and only then outputs 'True' or 'False'; in Step 4 it diagnoses why the previous keywords were insufficient before proposing a refined comma separated list. These changes preserve the BM25 backbone and add no extra retrieval overhead. Table 16 lists the modified prompts.

Table 17 compares the original ITERKEY (BM25) with its CoT variant on four QA benchmarks. Across models and tasks, the CoT variant fails to produce a consistent trend: NaturalQA

| Step | Prompt (CoT version) |
|---|---|
| Step 3: Answer Validation (CoT) | **System:** You are an assistant that verifies whether an answer is correct based on retrieved documents.
**User:** Question: { $q$ }
Answer: { $a$ }
Documents: { $Docs$ }
Let's think step by step.
**Chain of thought:** (Explain your reasoning step by step. Refer to the documents when relevant.)
**Conclusion:** True **or** False
*Do not output anything besides the above format.* |
| Step 4: Keyword Regeneration (CoT) | **System:** You refine keywords to improve document retrieval for BM25 search in the RAG framework.

**User:** The previous keywords failed to retrieve useful documents for the query: { $q$ }
Here are the previous keywords: { *prev_keywords* }
Let's think step by step. Check if they are too general, too specific, or missing important concepts.
Then, refine them to produce a more effective list of keywords for retrieving documents that contain the correct answer.
Output the keywords in the exact format: ["kw1", "kw2", ...]
Refined Keywords: |

Table 16: CoT enhanced prompts used in Step 3 (Answer Validation) and Step 4 (Keyword Regeneration).

| Setting | Model | Size (B) | Entity QA | HotpotQA | Natural QA | WebQA |
|---|---|---|---|---|---|---|
| IterKey (BM25) | Llama-3.1 | 8B | 62.0 | 49.7 | 50.2 | 53.3 |
| | Llama-3.1 | 70B | 63.5 | 54.9 | 53.8 | 57.7 |
| | Gemma-2 | 9B | 36.1 | 26.1 | 35.9 | 36.1 |
| | Phi-3.5-mini | 3.8B | 47.9 | 42.7 | 36.8 | 41.2 |
| **IterKey (BM25) + CoT** | Llama-3.1 | 8B | -1.6 60.4 | +1.1 50.8 | +3.0 53.2 | -1.1 52.2 |
| | Llama-3.1 | 70B | -0.3 63.2 | +0.5 55.4 | +1.6 55.4 | -2.1 55.6 |
| | Gemma-2 | 9B | -3.7 32.4 | -1.5 24.6 | -5.9 30.0 | -5.1 31.0 |
| | Phi-3.5-mini | 3.8B | +0.3 48.2 | -0.3 42.4 | -0.2 36.6 | -0.6 40.6 |

Table 17: Score changes when applying CoT to IterKey ( blue indicates improvement, red indicates decline).

improves modestly by up to +1.6, whereas EntityQA, HotpotQA, and WebQA show declines of up to -2.8. These results indicate that the clearly defined instructions in each step of ITERKEY are sufficient to leverage the LLM's capabilities, making additional CoT prompting unnecessary in this BM25 centric setting.

# E  Keyword Numbers Distribution in ITERKEY

Figure 3 shows the average and variance of the number of keywords generated by each model across all tasks and iterations. Most models generate between 9 and 12 keywords, with a maximum of under 30. This distribution helps analyze the relationship between keyword generation and performance in ITERKEY. Despite the size difference, both Llama-3.1-8B and Llama-3.1-70B exhibit similar trends, peaking at 8-10 keywords with nearly identical distribution shapes. On the other hand, Gemma-2 model generates the highest average number of keywords (12.39) compared to the other models, with a more concentrated distribution. However, as discussed in Section 3.2, 4.1 its recall rate and EM score are lower, particularly in keyword generation and iterative refinement, where it underperforms. This suggests that generating more keywords doesn't necessarily improve accuracy, likely due to issues in the validation step identified earlier. Phi-3.5-mini model shows a similar

keyword generation distribution to Llama-3.1-8B, but differences in recall rates and EM scores indicate variations in keyword quality. This underscores that performance depends not just on the number of keywords but on their quality as well.

Figures 4, 5, 6, 7 and Table 20 show how keyword generation evolves over five iterations for each model. Llama-3.1-8B consistently improves by reducing average keyword count and variance, contributing to strong performance. Llama-3.1-70B maintains stable keyword generation with slightly higher averages and increasing variance, reflecting flexibility without a notable drop in quality. In contrast, Gemma-2 generates the most keywords (mean: 12.8) but shows limited refinement and low variance, possibly limiting dynamic adjustment and lowering recall and EM scores. Phi-3.5-mini starts with a high keyword count but drops sharply, with rising variance, yet shows little performance improvement, likely due to inefficient refinement. These results suggest that keyword refinement is key, with Llama models showing a strong link to performance, while Gemma-2 and Phi-3.5-mini struggle with dynamic adjustment, potentially affecting their results.

## F  Impact of Chunk Size on ITERKEY

This study examines the impact of different chunk sizes and Top-k retrieved documents on model performance in QA tasks. Several key trends emerged from the results, as shown in Table 21. In these additional experiments, we used the December 2018 Wikipedia dump as the retrieval corpus for all datasets (Izacard et al., 2024). The text was divided into segments of tokens, with an overlap of 50 tokens at the beginning and end of each chunk to preserve context.

In the 'Chunk256' 'Top3' setting, the Llama-3.1-8B model outperformed other models across all QA tasks, achieving the best performance. This indicates that retrieving Top3' documents improves the model's accuracy over Top1'. On the other hand, when only the 'Top1' document is provided, the larger Llama-3.1-70B model generally outperformed the Llama-3.1-8B model, especially in tasks like EntityQA and WebQA. This suggests that the 70B model has the capacity to leverage a smaller set of relevant information more effectively, maintaining high accuracy with fewer retrieved documents.

Furthermore, when comparing chunk sizes, the results showed that increasing the chunk size from 'Chunk256' to 'Chunk512' led to a decline in performance, particularly in the 'Top3' configuration. For example, in the 'Chunk512' 'Top3' setting, both the Llama-3.1-8B and Llama-3.1-70B models exhibited significant drops in performance across all tasks, with notable declines in EntityQA and WebQA. This suggests that larger chunk sizes exceed the model's optimal token processing capacity, causing difficulties in handling the larger input context, and consequently leading to performance degradation.

Regarding model size and information efficiency, the performance difference between Llama-3.1-8B and Llama-3.1-70B reveals an interesting tradeoff. While the 70B model benefits from a larger parameter space, allowing it to excel with fewer documents ('Top1'), the 8B model makes better use of multiple documents ('Top3'). This indicates that smaller models can compensate for their parameter limitations by utilizing more retrieved information, whereas larger models can achieve comparable or better results with minimal inputs.

| Method | Model | Size (B) | Entity QA | HotpotQA | Natural QA | WebQA |
|---|---|---|---|---|---|---|
| **Vanilla** | Llama-3.1 | 8B | 33.6 | 31.2 | 40.6 | 53.4 |
| | Llama-3.1 | 70B | 45.2 | 41.4 | 46.0 | 54.0 |
| | Gemma-2 | 9B | 10.6 | 11.6 | 9.2 | 20.8 |
| | Phi-3.5-mini | 3.8B | 24.6 | 25.4 | 25.8 | 44.0 |
| **RAG (BM25)** | Llama-3.1 | 8B | 54.0 | 47.0 | 44.8 | 51.4 |
| | Llama-3.1 | 70B | 54.6 | 46.2 | 43.4 | 47.4 |
| | Gemma-2 | 9B | 47.9 | 39.6 | 33.2 | 41.6 |
| | Phi-3.5-mini | 3.8B | 48.2 | 42.2 | 32.6 | 40.2 |
| **RAG (E5)** | Llama-3.1 | 8B | 50.9 | 46.7 | 47.2 | 48.2 |
| | Llama-3.1 | 70B | 56.5 | 52.0 | 46.4 | 45.0 |
| | Gemma-2 | 9B | 52.2 | 40.8 | 41.5 | 41.8 |
| | Phi-3.5-mini | 3.8B | 50.2 | 44.6 | 37.0 | 41.4 |
| **RAG (BGE)** | Llama-3.1 | 8B | 48.3 | 46.7 | 45.7 | 47.2 |
| | Llama-3.1 | 70B | 53.3 | 50.0 | 48.4 | 48.8 |
| | Gemma-2 | 9B | 49.2 | 36.5 | 41.6 | 42.8 |
| | Phi-3.5-mini | 3.8B | 46.2 | 43.9 | 36.0 | 44.8 |
| **RAG (Contriever)** | Llama-3.1 | 8B | 47.4 | 47.7 | 43.6 | 48.2 |
| | Llama-3.1 | 70B | 51.0 | 51.0 | 47.4 | 48.8 |
| | Gemma-2 | 9B | 48.5 | 40.8 | 38.8 | 41.8 |
| | Phi-3.5-mini | 3.8B | 44.2 | 44.6 | 39.0 | 41.4 |
| **IterKey (BM25)** | Llama-3.1 | 8B | 62.0 | 49.7 | 50.2 | 53.3 |
| | Llama-3.1 | 70B | 63.5 | 54.9 | 53.8 | 57.7 |
| | Gemma-2 | 9B | 36.1 | 26.1 | 35.9 | 36.1 |
| | Phi-3.5-mini | 3.8B | 47.9 | 42.7 | 36.8 | 41.2 |

Table 18: 'Vanilla' uses no retrieval. 'RAG (BM25)', 'RAG (E5)', 'RAG (BGE)', and 'RAG (Contriever)' apply a single retrieval step based on the original query, using BM25, E5, BGE, and Contriever respectively. Our proposed 'IterKey (BM25)' iteratively generates and refines keywords, performing up to five retrieval iterations to optimize answer quality.

| Method | Model | Size (B) | Entity QA | HotpotQA | Natural QA | WebQA |
|---|---|---|---|---|---|---|
| **Vanilla** | Llama-3.1 | 8B | 33.6 | 31.2 | 40.6 | 53.4 |
| | Llama-3.1 | 70B | 45.2 | 41.4 | 46.0 | 54.0 |
| | Gemma-2 | 9B | 10.6 | 11.6 | 9.2 | 20.8 |
| | Phi-3.5-mini | 3.8B | 24.6 | 25.4 | 25.8 | 44.0 |
| **RAG (BM25)** | Llama-3.1 | 8B | 54.0 | 47.0 | 44.8 | 51.4 |
| | Llama-3.1 | 70B | 54.6 | 46.2 | 43.4 | 47.4 |
| | Gemma-2 | 9B | 47.9 | 39.6 | 33.2 | 41.6 |
| | Phi-3.5-mini | 3.8B | 48.2 | 42.2 | 32.6 | 40.2 |
| **ITRG Refine (E5)** | Llama-3.1 | 8B | 60.6 | 53.4 | 53.6 | 56.2 |
| | Llama-3.1 | 70B | 57.1 | 52.9 | 53.3 | 51.6 |
| | Gemma-2 | 9B | 54.2 | 47.6 | 47.4 | 48.5 |
| | Phi-3.5-mini | 3.8B | 54.3 | 47.1 | 36.2 | 45.6 |
| **ITRG Refresh (E5)** | Llama-3.1 | 8B | 58.5 | 49.2 | 52.2 | 56.3 |
| | Llama-3.1 | 70B | 60.1 | 53.2 | 55.7 | 52.2 |
| | Gemma-2 | 9B | 50.2 | 40.8 | 45.9 | 49.3 |
| | Phi-3.5-mini | 3.8B | 49.3 | 43.9 | 36.0 | 41.4 |
| **IterKey (BM25)** | Llama-3.1 | 8B | 62.0 | 49.7 | 50.2 | 53.3 |
| | Llama-3.1 | 70B | 63.5 | 54.9 | 53.8 | 57.7 |
| | Gemma-2 | 9B | 36.1 | 26.1 | 35.9 | 36.1 |
| | Phi-3.5-mini | 3.8B | 47.9 | 42.7 | 36.8 | 41.2 |

Table 19: 'Vanilla' does not use retrieval. 'RAG (BM25)' apply a single retrieval step based on the original query, using BM25. 'ITRG Refine (E5)' is a method that utilizes E5 for iterative refinement. 'ITRG Refresh (E5)' also uses E5, performing refinement based on query refreshment. The proposed method, 'IterKey (BM25)', iteratively generates and refines keywords, performing up to five retrieval iterations to optimize answer quality.

| Model | Iteration 1 | | Iteration 2 | | Iteration 3 | | Iteration 4 | | Iteration 5 | |
|---|---|---|---|---|---|---|---|---|---|---|
| | Mean | Var | Mean | Var | Mean | Var | Mean | Var | Mean | Var |
| Llama-3.1 8B | 11.2 | 4.1 | 9.3 | 3.9 | 8.5 | 4.0 | 8.1 | 3.8 | 8.1 | 3.9 |
| Llama-3.1 70B | 11.2 | 4.1 | 10.3 | 4.8 | 10.3 | 4.8 | 10.6 | 4.8 | 10.7 | 4.9 |
| Gemma-2 9B | 12.8 | 2.5 | 12.4 | 3.3 | 12.3 | 3.6 | 12.3 | 3.6 | 12.2 | 3.7 |
| Phi-3.5-mini 3.8B | 13.8 | 3.4 | 9.4 | 3.6 | 8.6 | 4.0 | 8.4 | 4.0 | 8.6 | 4.0 |

Table 20: Results of different models across iterations, displaying both the mean and variance.

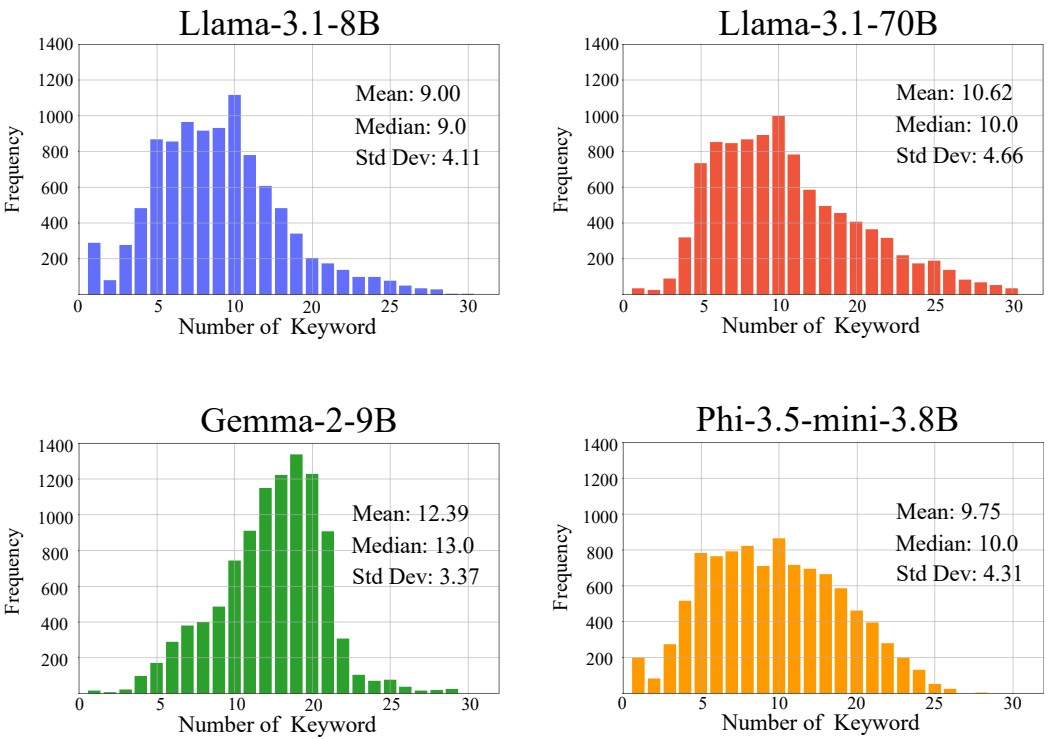

Figure 3: The distribution of keywords generated by each LLMs.

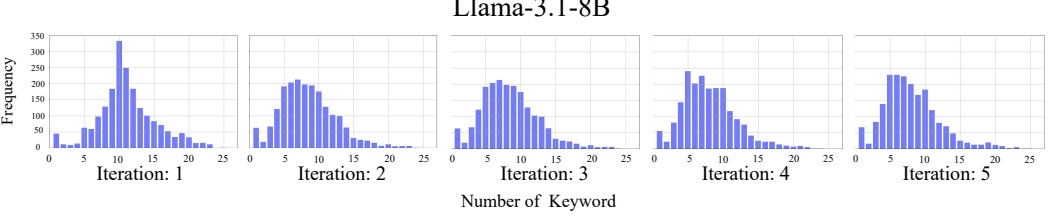

Figure 4: The distribution of the number of keywords per iteration in LLama 3.1-8B.

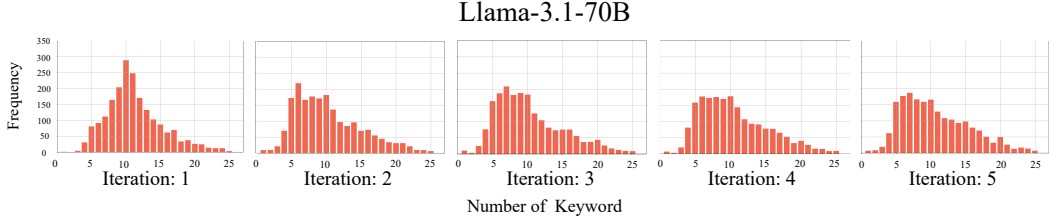

Figure 5: The distribution of the number of keywords per iteration in LLama 3.1-70B.

Gemma-2-9B

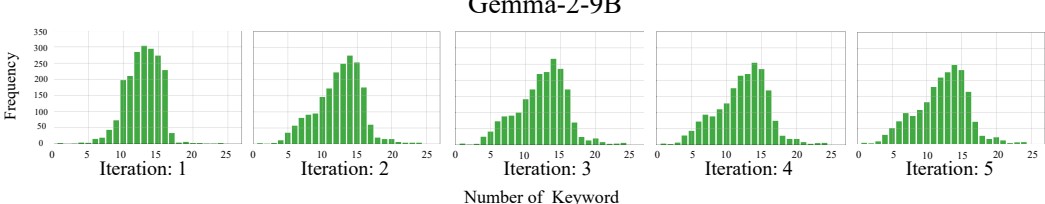

Figure 6: The distribution of the number of keywords per iteration in Gemma-2.

Phi-3.5-mini-3.8B

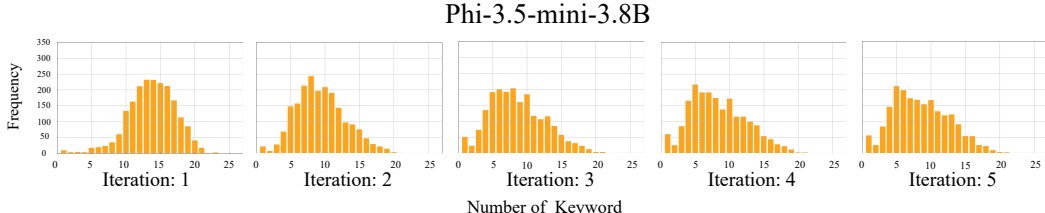

Figure 7: The distribution of the number of keywords per iteration in Phi-3.5-mini.

| Method | Model | Size (B) | **Entity QA** | **HotpotQA** | **Natural QA** | **WebQA** |
|---|---|---|---|---|---|---|
| | Llama-3.1 | 8B | 46.2 | 41.2 | 47.6 | 56.4 |
| Chunk256 Top1 | Llama-3.1 | 70B | 53.1 | 44.4 | 49.0 | 55.0 |
| | Gemma-2 | 9B | 24.0 | 7.2 | 10.0 | 18.8 |
| | Phi-3.5-mini | 3.8B | 34.2 | 28.0 | 28.8 | 43.2 |
| | Llama-3.1 | 8B | **53.6** | **47.8** | **54.8** | **59.8** |
| Chunk256 Top3 | Llama-3.1 | 70B | 52.0 | 47.0 | 52.6 | 55.6 |
| | Gemma-2 | 9B | 7.2 | 7.6 | 8.1 | 13.8 |
| | Phi-3.5-mini | 3.8B | 41.8 | 33.6 | 35.2 | 49.4 |
| | Llama-3.1 | 8B | 49.6 | 42.2 | 47.2 | 55.2 |
| Chunk512 Top1 | Llama-3.1 | 70B | 48.7 | 44.4 | 50.4 | 54.3 |
| | Gemma-2 | 9B | 10.6 | 7.2 | 10.0 | 12.0 |
| | Phi-3.5-mini | 3.8B | 35.2 | 30.0 | 31.9 | 46.1 |
| | Llama-3.1 | 8B | 22.6 | 15.6 | 23.5 | 29.2 |
| Chunk512 Top3 | Llama-3.1 | 70B | 29.7 | 23.4 | 33.4 | 37.3 |
| | Gemma-2 | 9B | 0.6 | 1.6 | 1.0 | 2.1 |
| | Phi-3.5-mini | 3.8B | 16.2 | 11.0 | 17.6 | 24.6 |

Table 21: Comparison of model performance on QA tasks with different chunk sizes and Top retrieved documents provided to the model during RAG. For example, with 'Chunk256' and 'Top1'

**Question:** Where does the PATH train stop in Newark?
**Answer:** Newark Penn Station

| Iteration | Keywords | Model Output | Validation Output | Correct? |
|---|---|---|---|---|
| 1 | PATH train, Newark, stops, transportation, stations, Penn Station, downtown Newark, train stations, commute, New Jersey, NJ Transit, rail, Newark PATH stops, PATH Newark stops, Newark train stops | Answer: Newark Liberty International Airport Station | False | False |
| 2 | Newark, New Jersey, NJ Transit, Penn Station, Newark Penn Station, downtown Newark, transportation, train stations, rail, commute | The PATH train stops at Newark Penn Station. | True | True |

Table 22: Case 1: Correct answer is reached in the second iteration.

**Question:** Who wrote and performed "I Can Only Imagine"?
**Answer:** Bart Millard

| Iteration | Keywords | Model Output | Validation Output | Correct? |
|---|---|---|---|---|
| 1 | I Can Only Imagine, MercyMe, Bart Millard, Christian Song, Christian Music Artist, Lead Vocalist, Songwriter, Christian Band, Contemporary Christian Music | Bart Millard | False | False |
| 2 | I Can Only Imagine, Bart Millard, MercyMe, Christian Song, Christian Music Artist, Lead Vocalist, Songwriter, Christian Band, Contemporary Christian Music | Bart Millard | False | False |
| 3 | I Can Only Imagine, MercyMe, Bart Millard, Christian Song, Worship Music, Christian Music Artist, Lead Vocalist, Songwriter, Christian Band | Bart Millard wrote and performed "I Can Only Imagine". | True | True |

Table 23: Case 2: Correct Answer with False Validation.

**Question:** What prolific English engineer, who died on 15 December 1857, contributed to the advent of the hot air engine by way of their open cycle devices?
**Answer:** Sir George Cayley

| Iteration | Keywords | Model Output | Validation Output | Correct? |
|---|---|---|---|---|
| 1 | English engineer, hot air engine, open cycle, December 1857 | Sir George Cayley | False | False |
| 2 | English engineer, hot air engine, open cycle, 15 December 1857, inventor, hot air engine inventor | Sir George Cayley | False | False |
| 3 | English engineer, hot air engine, open cycle, inventor, 15 December 1857, British engineer, thermodynamic cycle | Sir George Cayley | False | False |
| 4 | English engineer, hot air engine, open cycle, thermodynamic cycle, 15 December 1857, British engineer, inventor, heat engine | Sir George Cayley | False | False |
| 5 | English engineer, hot air engine, open cycle, thermodynamic cycle, 1857, British engineer, inventor, heat engine, Stirling engine, Robert Stirling | Sir George Cayley | False | False |

Table 24: Case 3: Model fails to verify the correct answer despite arriving at it.

