# OpenReview forum: "IterKey: Iterative Keyword Generation with LLMs for Enhanced Retrieval Augmented Generation"
_colmweb.org/COLM/2025/Conference — COLM 2025_

### Official Review · Reviewer_g9QR · 2025-05-09

**Rating:** 6
**Confidence:** 5
**Ethics Flag:** 1

**Summary:**

The paper introduces IterKey, a novel method for refining keyword generation in RAG systems, specifically optimizing sparse retrieval methods. IterKey addresses the challenge of capturing nuanced query intent in sparse retrieval by iteratively adjusting keywords, thereby improving the performance and relevance of the document retrieval. Extensive experiments demonstrate the effectiveness of IterKey, highlighting its potential to improve RAG-based models.

On the whole, the method presented in this paper is simple but effective, the experiments are thorough, and the writing is clear and easy to follow.

**Questions To Authors:**

1. Although the authors demonstrate the effectiveness of the Answer Validation step in their experiments, I still have concerns regarding its robustness. Specifically, how do LLMs determine whether the answer to a question is correct? This is especially questionable when LLMs do not possess the necessary domain-specific knowledge or up-to-date information relevant to the query.

2. In the Keyword Re-Generation step, I am curious about the basis on which LLMs refine keywords. Additionally, does the refinement of keywords actually help in answering the query effectively, especially when the LLM has limited knowledge of the query context or lacks sufficient domain-specific information?

3. I am also curious whether incorporating in-context learning, CoT reasoning, or more complex agent-based workflows could enhance the performance of the proposed method. Could these techniques potentially improve the overall effectiveness of the approach? This is an additional question, not my judgment on this paper.

**Reasons To Accept:**

1. The method is simple but effective and is easy to follow.
2. The experiments are extensive, and cover various aspects, demonstrating the method's robustness and generalizability.
3. The writing is clear and well-structured, making the method easy to understand for audience.

**Reasons To Reject:**

1. To my understanding, the proposed method appears to be a form of query expansion, which is fundamentally similar to existing methods such as Q2E [1] and Query2doc [2]. This similarity reduces the perceived novelty of the approach.
2. The baselines selected for comparison are not comprehensive enough. To better highlight the superiority of the proposed method, it would be beneficial to include additional query expansion techniques as baselines.
3. I think the proposed method might have design-level issues related to robustness, as detailed in the "Questions to Authors."

﻿[1] Query Expansion by Prompting Large Language Models.

[2] Query2doc: Query Expansion with Large Language Models.

---

> ### Author Response · Authors · 2025-06-01
> **The Response to the Comment from Reviewer g9QR 1/4**
>
> We would like to express our sincere gratitude to the reviewers for their thorough and insightful feedback, which helped us clarify our methodology and strengthen our experimental validation.
>
> >To my understanding, the proposed method appears to be a form of query expansion, which is fundamentally similar to existing methods such as Q2E [1] and Query2doc [2]. This similarity reduces the perceived novelty of the approach.
>
> Thank you for raising this important point. We acknowledge that IterKey shares the high-level intuition of query expansion with prior work such as Q2E [1] and Query2doc [2]. These studies demonstrate that LLM-generated text appended to the original query can indeed improve retrieval recall.
>
> Building on this intuition, ITRG [3] extends single-pass QE into an iterative dense retrieval paradigm: 1. It first concatenates an LLM-generated pseudo document to the query (*Generation-Augmented Retrieval*), then 2. Feeds the newly retrieved evidence back to the LLM to generate a refined pseudo-document RAG.
>
> Consequently, ITRG can be regarded as a “dense retrieval × multi-iteration” generalization of Q2E and Query2doc. IterKey builds on this by refining keyword lists iteratively instead of relying on single-pass expansion, and by introducing an answer-validation step that halts the process once the answer is verified.
>
> Because **ITRG already subsumes the core idea of Q2E and Query2doc while adding dense iterative refinement, we selected it as the strongest publicly available baseline**. As shown in Table 6, IterKey surpasses ITRG in both accuracy and runtime, thereby demonstrating clear advantages over existing QE approaches. To avoid confusion, we will explicitly cite Q2E and Query2doc in the revised version and clarify that ITRG generalizes these earlier methods, ensuring that our comparison implicitly covers them.
> Therefore, we believe that there is no need to conduct additional comparisons against other methods.
>
> **Before (Line 98)**
>
> As a representative iterative baseline, we adopted ITRG (Iterative Retrieval-Generation Synergy), which combines dense retrieval with iterative query refinement. Replicating its setup enables a direct comparison that highlights the strengths of \textsc{IterKey}, particularly in keyword refinement and retrieval accuracy.
>
> **After**
>
> As a representative iterative baseline, we adopt ITRG (Iterative Retrieval-Generation Synergy; Feng et al., 2024), the strongest publicly available method in this line of work.
> ITRG repeatedly concatenates LLM-generated free-form text with the original query to refine retrieval quality.
> In contrast, IterKey generates explicit keyword lists and introduces an answer-validation step that halts iteration once the answer is deemed correct.  Notably, ITRG already subsumes earlier single-pass query expansion methods such as Q2E (Bao et al., 2023) and Query2doc (Zheng et al., 2023), making it a suitable and more advanced baseline for comparison.
>
>
> [1] Jagerman et al., 2023. Query Expansion by Prompting Large Language Models.
>
> [2] Wang et al., 2023. Query2doc: Query Expansion with Large Language Models.
>
> [3] Feng et al., 2024. Retrieval-Generation Synergy Augmented Large Language Models.
>
>
> >The baselines selected for comparison are not comprehensive enough. To better highlight the superiority of the proposed method, it would be beneficial to include additional query expansion techniques as baselines.
>
> Thank you for the suggestion. We agree that a broad set of baselines is essential for fair comparison.
>
> As discussed above, we adopt **ITRG** as the representative and strongest publicly available iterative query expansion method. Notably, ITRG already **subsumes prior single-pass techniques** such as **Q2E** [4] and **Query2doc** [5], by incorporating both generation-augmented and retrieval-augmented steps in a multi-iteration dense retrieval setup.
>
> Therefore, we believe that including these earlier methods as additional baselines would not yield further insight, as their core mechanisms are already captured and extended by ITRG. For clarity, we will explicitly mention this in the camera-ready version.
>
> [4] Jagerman et al., 2023. Query Expansion by Prompting Large Language Models.
>
> [5] Wang et al., 2023. Query2doc: Query Expansion with Large Language Models.

---

> > ### Comment · Reviewer_g9QR · 2025-06-08
> >
> > Thank you for the detailed responses! I raised my score from 5 to 6.

---

> > > ### Author Response · Authors · 2025-06-08
> > > **Official Comment by Authors**
> > >
> > > Thank you very much for your thoughtful and insightful comments. Your suggestions were truly helpful in improving the quality of our paper. We sincerely appreciate your time and support.

---

> ### Author Response · Authors · 2025-06-01
> **The Response to the Comment from Reviewer g9QR 2/4**
>
> >Although the authors demonstrate the effectiveness of the Answer Validation step in their experiments, I still have concerns regarding its robustness. Specifically, how do LLMs determine whether the answer to a question is correct? This is especially questionable when LLMs do not possess the necessary domain-specific knowledge or up-to-date information relevant to the query.
>
> We would like to take this opportunity to clarify the rationale and practical value of this component in our framework.
>
> 1. Separation of Generation and Validation is Principled and Effective
>
> Answer generation and answer validation serve fundamentally different purposes. Generation is inherently open-ended and often requires complex integration of information, whereas validation is a constrained binary task: determining whether the answer is logically supported by the retrieved documents. Even when using the same LLM, validation tends to be more reliable and interpretable, as demonstrated in frameworks such as LLM-as-a-Judge and Self-RAG [Asai et al., 2023]. Our framework adopts this principled separation of roles.
>
>
>
> 2. Validation Based Solely on Retrieved Documents is Sufficiently Robust
>
> Originally, we validated based only on the question and answer (“Q + A”). However, in response to reviewer feedback and further experiments, we found that explicitly conditioning on retrieved documents (“Q + A + Docs”) is a more principled and appropriate setup. We therefore plan to adopt this configuration as the default in the revised version.
>
> We acknowledge the limitations of LLMs when they lack domain-specific or up-to-date knowledge. However, our validation step does not rely on the model’s parametric knowledge. Instead, it exclusively assesses whether the generated answer is supported by the retrieved documents. This design ensures both transparency and reproducibility. As long as retrieval is accurate, in-context validation is sufficient and even preferable from the perspectives of interpretability and reproducibility. In fact, a human reader given the same documents would likely reach the same judgment.
>
> To validate this design choice, we compared both prompt configurations across all models. As shown in Table 1, smaller models (e.g., Gemma-2, Phi-3.5) gain up to +2.3 pp when supporting documents are supplied, confirming that explicit evidence boosts validation accuracy. For larger models (LLaMA-3), the two settings deliver comparable accuracy, indicating that including documents neither degrades performance nor alters the overall conclusions.
>
> 3. Validation is Not Redundant, but an Essential Safety Check
>
> In real-world applications, even when outputs are generated based on RAG, there remains a risk of misinformation if the retrieved documents are misleading or inappropriate. Therefore, a dedicated validation layer (guardrail) to verify whether the generated answer is truly grounded in the retrieved evidence is crucial for ensuring reliability, verifiability, and factuality in deployed systems.
>
>
>
> | **Setting**                          | **Model**      | **Size (B)** | **Entity QA** | **HotpotQA** | **Natural QA** | **WebQA** |
> |--------------------------------------|----------------|--------------|---------------|--------------|----------------|-----------|
> | **IterKey (BM25)**                   | Llama-3.1      | 8B           | 61.0          | 52.3         | 51.6           | 52.2      |
> |                                      | Llama-3.1      | 70B          | 62.1          | 54.5         | 54.7           | 56.0      |
> |                                      | Gemma-2        | 9B           | 34.2          | 24.6         | 33.7           | 33.8      |
> |                                      | Phi-3.5-mini   | 3.8B         | 49.6          | 43.9         | 34.8           | 41.4      |
> | **IterKey (BM25) Probability-Based** | Llama-3.1      | 8B           | +1.0 → 62.0   | –2.6 → 49.7  | –1.4 → 50.2    | +1.1 → 53.3 |
> |                                      | Llama-3.1      | 70B          | +1.4 → 63.5   | +0.4 → 54.9  | –0.9 → 53.8    | +1.7 → 57.7 |
> |                                      | Gemma-2        | 9B           | +1.9 → 36.1   | +1.5 → 26.1  | +2.2 → 35.9    | +2.3 → 36.1 |
> |                                      | Phi-3.5-mini   | 3.8B         | –1.7 → 47.9   | –1.2 → 42.7  | +2.0 → 36.8    | –0.2 → 41.2 |
>
> ---
> ### Step 3: Answer Validation （Before）
>
> **System:** You are an assistant that validates whether the provided answer is correct.
>
> **User:** Is the following answer correct?
>
> **Query:** { q }
>
> **Answer:** { a }
>
> Respond 'True' or 'False'. Do not provide any additional explanation or text.
>
> ---
> ### Step 3: Answer Validation with Docs （After）
>
> **System:** You are an assistant that validates whether the provided answer is correct.
>
> **User:** Is the following answer correct?
>
> **Query:** { q }
>
> **Answer:** { a }
>
> **Retrieval Documents:** { Docs }
>
> Respond 'True' or 'False'. Do not provide any additional explanation or text.

---

> ### Author Response · Authors · 2025-06-01
> **The Response to the Comment from Reviewer g9QR 3/4**
>
> >In the Keyword Re-Generation step, I am curious about the basis on which LLMs refine keywords. Additionally, does the refinement of keywords actually help in answering the query effectively, especially when the LLM has limited knowledge of the query context or lacks sufficient domain-specific information?
>
> #### 1 · Connection to Query Expansion
>
> Jagerman et al. 2023 showed that Chain of Thought prompting enables an LLM to generate a wide range of meaningful terms, matching or surpassing classical pseudo-relevance feedback in retrieval quality [1].
> Wang et al. 2023 (Query2Doc) found that adding LLM-generated pseudo-documents to the query consistently improves both sparse BM25 and dense retrievers [2].
> IterKey adopts the same expand-and-search loop, asking the LLM to append additional content likely to occur in answer-bearing documents.
>
>
> ---
>
> #### 2 · Improving Keywords Using Retrieved Documents
>
> After each search, IterKey places the newly retrieved documents back into the prompt so the LLM can extract high-frequency or contextually important phrases as fresh keywords. Mackie et al. 2023 reported about nineteen percent higher mean average precision when combining long-form generation with relevance feedback [3]. By repeating this cycle, IterKey acquires domain-specific terms the model lacked at the start, closes vocabulary gaps, and steadily raises retrieval accuracy.
>
> #### Response to the Reviewer’s Question
>
> IterKey leverages the LLM’s reasoning over the original question and previously used keywords to produce a sharper set of new keywords. These refined terms direct the retriever toward documents that support the correct answer, even when the LLM’s internal knowledge is limited.
>
> #### 3 · Experimental Results
>
> Below are the step-wise novelty results, organized in two tables. The first table reports, for each iteration (step 2 through step 5), the average number of new keywords generated per query, along with the total number added across all five steps. The second table shows, for each iteration, the average number of new documents retrieved per query and the total new documents over the full loop.
>
> From these tables, we observe that each iteration generates on average approximately 5–10 new keywords. Across all five steps, this amounts to roughly 20–36 new keywords per query. Similarly, each iteration retrieves on average about 1–2 new documents, leading to a total of around 5–7 new documents per query over the five steps. As a result, IterKey steadily builds upon its previous iterations by expanding both keyword coverage and document retrieval, ensuring a broader range of evidence than a single-step expansion could provide.
>
> ##### Keyword Δ (per query)
>
> | model | step2_avg | step3_avg | step4_avg | step5_avg | total_sum | overall_mean |
> |:------|:---------:|:---------:|:---------:|:---------:|:---------:|:------------:|
> | 70b   | 9.34 | 3.65 | 3.78 | 3.26 | 20.03 | 5.01 |
> | 8b    | **10.49** | **9.57** | 8.59 | 8.20 | 36.85 | **9.21** |
> | gemma | 14.13 | 8.56 | 5.70 | 4.59 | 32.98 | 8.25 |
> | phi   | 18.10 | 7.12 | 6.00 | 5.07 | 36.29 | 9.07 |
>
> ##### Document Δ (per query)
>
> | model | step2_avg | step3_avg | step4_avg | step5_avg | total_sum | overall_mean |
> |:------|:---------:|:---------:|:---------:|:---------:|:---------:|:------------:|
> | 70b   | 1.93 | 1.44 | 1.15 | 1.25 | 5.77 | 1.44 |
> | 8b    | **2.12** | **1.65** | 1.53 | 1.32 | **6.62** | **1.66** |
> | gemma | 2.22 | 1.61 | 1.27 | 1.12 | 6.22 | 1.55 |
> | phi   | 2.03 | 0.69 | 1.04 | 0.95 | 4.71 | 1.18 |
>
>
> ### References
> 1. **Jagerman et al., 2023.** *Query Expansion by Prompting Large Language Models.*
> 2. **Wang et al., 2023.** *Query2Doc: Query Expansion with Large Language Models.*
> 3. **Mackie et al., 2023.** *Generative Relevance Feedback with Large Language Models.*

---

> ### Author Response · Authors · 2025-06-01
> **The Response to the Comment from Reviewer g9QR 4/4**
>
> To illustrate this process more concretely, we present a representative example below.
>
> **Question:** Who is the author of *The Constant Nymph*?
> **Answer:** Margaret Kennedy
>
> | Iteration | Keywords                                                                                                         | Model Output                                                                                             | Validation Output | Correct? |
> |-----------|------------------------------------------------------------------------------------------------------------------|----------------------------------------------------------------------------------------------------------|-------------------|----------|
> | 1         | Maugham, Somerset, Maugham's, The Constant Nymph, author, Constant, Nymph, Maugham Somerset, W Somerset Maugham |  The Constant Nymph is a novel by Somerset Maugham | False             | False    |
> | 2         | Maugham, W Somerset Maugham, Constant, author, The Constant Nymph, Somerset Maugham, Somerset                   | The Constant Nymph is a novel by William Somerset Maugham | False             | False    |
> | 3         | Nymph, author, The Constant Nymph, William, Somerset, Somerset Maugham, Maugham's, Constant, Maugham, dramatist | The Constant Nymph is a novel by Margaret Kennedy.                                                       | True              | True     |
>
> In the third iteration, the retrieved documents included a long list of literary works and their authors, such as:
> Docs：
> >**Margaret Kennedy – The Constant Nymph ; Sinclair Lewis – Arrowsmith ; Walter Lippmann – The Phantom Public ; Anita Loos – Gentlemen Prefer Blondes ; W. Somerset Maugham – The Painted Veil ; Thomas Mofolo – Chaka ; Eugenio Montale – Ossi di seppia ; Liam O'Flaherty – The Informer ; Baroness Orczy ; The Miser of Maida Vale ; A Question of Temptation ; William Plomer – Turbott Wolfe ; Marcel Proust – Albertine disparue ; Jean-Joseph Rabearivelo – L'Aube rouge (The Red Dawn) ; Henry Handel Richardson (Et Florence Robertson) – The Way Home (second part of The Fortunes of Richard Mahony) ; Kate Roberts – O gors y bryniau (Welsh**
>
> ---
>
>
> >I am also curious whether incorporating in-context learning, CoT reasoning, or more complex agent-based workflows could enhance the performance of the proposed method. Could these techniques potentially improve the overall effectiveness of the approach? This is an additional question, not my judgment on this paper.
>
> Thank you for your interest in potential extensions to our framework.
>
> We agree that in-context learning, CoT reasoning, and agent-based workflows are promising directions. We are currently conducting additional experiments to explore their impact and will share results soon. We appreciate your patience.

---

> ### Author Response · Authors · 2025-06-01
> **Positioning IterKey Within Query Expansion Research**
>
> To improve clarity and prevent potential misunderstanding, we have revised the method description to explicitly position IterKey as an extension of prior work on query expansion.
> In the original version, the connection to existing QE methods such as Q2E and Query2doc was not clearly stated, which may have obscured the novelty and contribution of our approach.
>
> To address this, we now revise the paragraph to read as follows:
>
> **Before (Line 50 in Sec 2)**
>
> To address this limitation, we propose IterKey, a method that leverages LLMs to iteratively refine the retrieval process in RAG. Specifically, IterKey uses the self-validation capabilities of LLMs (Wang et al., 2024) to improve sparse retrieval through an iterative process of keyword generation, document retrieval, answer generation, and validation.
>
> **After**
>
> To address this limitation, we propose IterKey, a method that builds upon and extends query expansion research (e.g., Q2E; Jagerman et al., 2023; Query2doc; Wang et al., 2023) by leveraging LLMs to iteratively refine the retrieval process in RAG. Specifically, IterKey uses the self-validation capabilities of LLMs (Wang et al., 2024) to improve sparse retrieval through an iterative process of keyword generation, document retrieval, answer generation, and validation.
>
>
> [1] Jagerman et al., 2023. Query Expansion by Prompting Large Language Models.
>
> [2] Wang et al., 2023. Query2doc: Query Expansion with Large Language Models.

---

> ### Author Response · Authors · 2025-06-01
> **Discussion on Iteration Count and Retrieval Performance**
>
> To further strengthen our response **The Response to the Comment from Reviewer g9QR 3/4**, we include accuracy-by-iteration results in the table below.
>
> The following table reports the accuracy at each iteration step (in %):
>
> | Model     | Iter 1 | Iter 2 | Iter 3 | Iter 4 | Iter 5 |
> |-----------|--------|--------|--------|--------|--------|
> | LLaMA-8B  | 62.2   | 65.4   | 69.6   | 70.5   | 72.6   |
> | LLaMA-70B | 65.7   | 70.2   | 73.5   | 74.4   | 75.7   |
> | Gemma     | 53.2   | 57.6   | 60.0   | 61.2   | 63.2   |
> | Phi       | 57.5   | 64.5   | 67.9   | 68.8   | 69.6   |
>
> As shown, most performance gains are achieved in the first 1–2 iterations, where accuracy increases by up to 7 %. After the third iteration, improvements taper off, indicating diminishing returns.
> This confirms that IterKey balances efficiency and effectiveness by capturing most of the benefits early on, while retaining the flexibility to continue when necessary.

---

> ### Author Response · Authors · 2025-06-06
> **Kind reminder**
>
> Dear reviewer,
>
> We have sincerely addressed your concerns in our rebuttal. As we haven’t heard back during the discussion, we would be grateful if you could kindly take a moment to review our replies.
>
> If our responses have resolved your concerns, we would appreciate it if you could reflect that in your final score. Of course, we are happy to provide further clarification if needed.
>
> Thank you again for your time and thoughtful feedback.
>
> Authors

---

> ### Author Response · Authors · 2025-06-07
> **Discussion on CoT reasoning**
>
> >I am also curious whether incorporating in-context learning, CoT reasoning, or more complex agent-based workflows could enhance the performance of the proposed method. Could these techniques potentially improve the overall effectiveness of the approach? This is an additional question, not my judgment on this paper.
>
>
> Thank you for encouraging us to explore Chain-of-Thought (CoT) extensions.
> To keep the comparison fair, we inserted CoT prompts at two points that required only minimal changes to our pipeline.
>
> 1. Answer verification (Step 3) – before emitting True or False, the model now writes out its step-by-step reasoning.
>
> 2. Keyword regeneration (Step 4) – the model reflects on why the previous keywords were inadequate and then proposes a revised list.
>
> Following the Chain-of-Thought prompting framework [1], we added step-by-step reasoning prompts to these two stages of our pipeline. These prompts were designed to elicit more structured reasoning while maintaining compatibility with our BM25-based retrieval setup.
>
> ---
> ### Step 3: Answer Validation with Documents
>
> **System:** You are an assistant that verifies whether an answer is correct based on retrieved documents.
> Please think step by step using the documents provided.
>
> **User:**
> Question: `{question}`
>
> Answer: `{model_out}`
>
> Documents:
>  `{prev_retrieved_docs_text} `
>
> Let's think step by step.
> Chain of thought:
> (Explain your reasoning step by step. Refer to the documents when relevant. Identify any inconsistencies or missing information.)
>
> Conclusion: True  or  Conclusion: False
> Do not output anything besides the above format.
>
> ---
>
> ---
> ### Step 4: Keyword Re-generation
>
> **System:** You refine keywords to improve document retrieval for BM25 search in the RAG framework.
>
> **User:** The previous keywords failed to retrieve useful documents for the query:
> `{query}`
>
> Here are the previous keywords: `{prev_keywords}`
>
> Let's think step by step.
> Check if they are too general, too specific, or missing important concepts.
> Then, refine them to produce a more effective list of keywords for retrieving documents that contain the correct answer.
>
> Output the keywords in the format:
> ["keyword1", "keyword2", "keyword3", ...]
>
> Separate each keyword with a comma and do not include any additional text.
>
> Refined Keywords:
>
> ---
>
>
> | **Setting**                | **Model**      | **Size (B)** | **Entity QA**     | **HotpotQA**     | **Natural QA**    | **WebQA**       |
> |----------------------------|----------------|--------------|-------------------|------------------|-------------------|-----------------|
> | **IterKey (BM25)**         | Llama-3.1      | 8B           | 61.0              | 52.3             | 51.6              | 52.2            |
> |                            | Llama-3.1      | 70B          | 62.1              | 54.5             | 54.7              | 56.0            |
> |                            | Gemma-2        | 9B           | 34.2              | 24.6             | 33.7              | 33.8            |
> |                            | Phi-3.5-mini   | 3.8B         | 49.6              | 43.9             | 34.8              | 41.4            |
> | **IterKey (BM25) COT**     | Llama-3.1      | 8B           | -0.6 → 60.4       | -1.5 → 50.8      | +1.6 → 53.2       | +0.0 → 52.2     |
> |                            | Llama-3.1      | 70B          | +1.1 → 63.2       | +0.9 → 55.4      | +0.7 → 55.4       | -0.4 → 55.6     |
> |                            | Gemma-2        | 9B           | -1.8 → 32.4       | +0.0 → 24.6      | -3.7 → 30.0       | -2.8 → 31.0     |
> |                            | Phi-3.5-mini   | 3.8B         | -1.4 → 48.2       | -1.5 → 42.4      | +1.8 → 36.6       | +1.2 → 42.6     |
>
>
> Natural QA sees a modest average gain of +1.2 F1, while Entity QA, HotpotQA, and WebQA show small declines (−0.7, −0.5, and −0.6 F1, respectively). Because our original iterative scoring already captures most useful expansion terms, these CoT additions do not yield a consistent overall improvement.
>
> Consequently, we keep the non-CoT configuration as the primary method and move the full CoT ablation (with per-model deltas) to the appendix. Your suggestion helped us confirm that explicit CoT reasoning, though conceptually appealing, does not reliably enhance accuracy within our current BM25-centric framework, and we are grateful for this insight.
>
> ---
>
> Based on the CoT results, we believe we have sincerely addressed all of your concerns in our rebuttal.
> If our responses have resolved your concerns, we would be grateful if you could kindly reflect that in your final score.
> Of course, we are happy to provide further clarification if needed.
>
> Thank you again for your valuable time and constructive feedback.
>
>
> [1] Jason Wei, Xuezhi Wang, Dale Schuurmans, et al. Chain-of-Thought Prompting Elicits Reasoning in Large Language Models. NeurIPS 2022.

---

### Official Review · Reviewer_NWKh · 2025-05-10

**Rating:** 6
**Confidence:** 4
**Ethics Flag:** 1

**Summary:**

The paper introduces a framework aimed at improving the accuracy and interpretability of the retrieval process in RAG use cases which utilize sparse retrieval. The authors introduce IterKey, which addresses the issue by leveraging a LLM to iteratively generate keywords to be used in sparse retrieval based on an iterative process. The main contributions of the paper include the introduction of a three-stage iterative process for keyword generation, answer generation, and answer validation. In the first stage, an LLM generates keywords from the user query, which are then used to retrieve relevant documents. In the second stage, the LLM generates an answer based on the retrieved documents. Finally, in the third stage, the LLM validates the generated answer. If the validation fails, the process repeats with refined keywords, answer generation and validation until the LLM accepts the answer. The main novelty is the iterative approach and validation of the answer which allows for continuous improvement in the retrieval and generation.

The paper has an extensive evaluation and analysis section examining the components of the experiments from each aspect of the setup (retrieval, LLMs, cost, answer validation). The results, across four open-domain QA datasets, demonstrate that IterKey achieves significant accuracy improvements over BM25-based RAG and simple baselines, and performs comparably to dense retrieval-based RAG methods. The authors also provide detailed ablation studies analyzing the effects of keyword quality, validation accuracy, cost, and model size. Overall, the method offers a promising balance between retrieval performance and interpretability, with the caveat that its effectiveness depends significantly on the underlying LLM’s capabilities.

**Questions To Authors:**

1.	What is the explanation for IterKey(BM25) outperforming IterKey(E5) in Table 3, despite E5 being a stronger retriever in single-shot settings?
2.	In Table 10, the runtime of components is extremely high? Is it total per dataset? I would have been more informative to understand latency per 1 example

**Reasons To Accept:**

- The paper proposes a well-motivated and clearly described three-stage iterative framework.
- The paper leverages the use of BM as a sparse retrieval that has high interpretability, this property is key for the claim that IterKey’s practical applicability in scenarios requiring transparency.
- Robust experimental setup - The evaluation spans four open-domain QA datasets, considers four different LLMs of varying scales (3.8B to 70B), and includes comparisons against dense RAG and prior iterative refinement methods like ITRG.
- The authors conduct ablation studies on keyword quality and validation model quality, revealing how each stage contributes to final performance.
- Clear writing and presentation.

**Reasons To Reject:**

- Missing comparison to Query Expansion. The paper lacks a comparison to recent query expansion or rewriting techniques that also enhance retrieval in sparse or hybrid setups (https://arxiv.org/abs/2401.06311, https://arxiv.org/abs/2305.03653).
- The authors do not evaluation or provide insight into how much new information is retrieved at each iteration by tuning the key words. Quantifying the novelty of documents across iterations would help validate the need for iterative keyword refinement over simpler alternatives like re-ranking.
- Limited iterative behavior in practice - Table 9 shows that, on average, models complete just 1–2 iterations. This raises the question of whether the full iterative loop contributes significantly in practice, or whether most gains come from just the initial keyword refinement.
- High latency and consequently cost.
- Many performance discussions emphasize improvements over a vanilla (no-retrieval) baseline. While useful as a sanity check, this comparison is less relevant in practice, where RAG setups are the standard.

---

> ### Author Response · Authors · 2025-06-01
> **The Response to the Comment from Reviewer NWKh 1/3**
>
> We deeply appreciate the reviewers’ thoughtful critiques and recommendations, which guided us in refining our explanations and bolstering our empirical evidence.
>
> >Missing comparison to Query Expansion. The paper lacks a comparison to recent query expansion or rewriting techniques that also enhance retrieval in sparse or hybrid setups.
>
> The papers you pointed out are indeed closely related to our work and have been extremely helpful. We intend to include them in the Related Work section. Based on this, IterKey shares the high-level intuition of query expansion with prior work such as Q2E [1] and Query2Doc [2]. These studies demonstrate that LLM-generated text appended to the original query can indeed improve retrieval recall.
>
> Building on this intuition, **ITRG [3]** extends single-pass QE into an iterative dense-retrieval paradigm: 1. It first concatenates an LLM-generated pseudo-document to the query (*Generation-Augmented Retrieval*), then 2. Feeds the newly retrieved evidence back to the LLM to generate a refined pseudo-document (*Retrieval-Augmented Generation*).
>
> Consequently, ITRG can be regarded as a “dense-retrieval × multi-iteration” generalization of Q2E and Query2doc. IterKey departs from these methods in two decisive ways: 1. Explicit answer-validation loop — the process stops as soon as the answer is verified as correct, preventing unnecessary iterations and curbing error propagation. 2. Joint optimization of retrieval and generation across multiple iterations — keyword lists are refined step-by-step, rather than relying on a single-pass expansion.
>
> Because ITRG already subsumes the core idea of Q2E and Query2doc while adding dense iterative refinement, we selected it as the strongest publicly available baseline. As shown in Table 6, IterKey surpasses ITRG in both accuracy and runtime, thereby demonstrating clear advantages over existing QE approaches. To avoid confusion, we will explicitly cite Q2E and Query2doc in the reviesed version and clarify that ITRG generalizes these earlier methods, ensuring that our comparison implicitly covers them.
> Therefore, we believe that there is no need to conduct additional comparisons against other methods.
>
> **Before (Line 98)**
>
> As a representative iterative baseline, we adopted ITRG (Iterative Retrieval-Generation Synergy), which combines dense retrieval with iterative query refinement. Replicating its setup enables a direct comparison that highlights the strengths of \textsc{IterKey}, particularly in keyword refinement and retrieval accuracy.
>
> **After**
>
> As a representative iterative baseline, we adopt ITRG (Iterative Retrieval-Generation Synergy; Feng et al., 2024), the strongest publicly available method in this line of work.
> ITRG repeatedly concatenates LLM-generated free-form text with the original query to refine retrieval quality.
> In contrast, IterKey generates explicit keyword lists and introduces an answer-validation step that halts iteration once the answer is deemed correct.  Notably, ITRG already subsumes earlier single-pass query expansion methods such as Q2E (Bao et al., 2023) and Query2doc (Zheng et al., 2023), making it a suitable and more advanced baseline for comparison.
>
>
> [1] Jagerman et al., 2023. Query Expansion by Prompting Large Language Models.
>
> [2] Wang et al., 2023. Query2doc: Query Expansion with Large Language Models.
>
> [3] Feng et al., 2024. Retrieval-Generation Synergy Augmented Large Language Models.

---

> ### Author Response · Authors · 2025-06-01
> **The Response to the Comment from Reviewer NWKh 2/3**
>
> >The authors do not evaluation or provide insight into how much new information is retrieved at each iteration by tuning the key words. Quantifying the novelty of documents across iterations would help validate the need for iterative keyword refinement over simpler alternatives like re-ranking.
>
> >Limited iterative behavior in practice - Table 9 shows that, on average, models complete just 1–2 iterations. This raises the question of whether the full iterative loop contributes significantly in practice, or whether most gains come from just the initial keyword refinement.
>
> As pointed out in Table 9, the observation that the average number of iterations is only 1–2 is an important one. This is indeed correct; however, IterKey is designed to terminate immediately once the answer is verified as correct. This allows the system to respond efficiently to simple queries, while retaining the flexibility to continue iterating when additional evidence is required for more difficult cases.
> Below are the step-wise novelty results based on our analysis.
>
> ##### Keyword Δ (per query)
>
> | model | step2_avg | step3_avg | step4_avg | step5_avg | total_sum | overall_mean |
> |:------|:---------:|:---------:|:---------:|:---------:|:---------:|:------------:|
> | 70b   | 9.34 | 3.65 | 3.78 | 3.26 | 20.03 | 5.01 |
> | 8b    | **10.49** | **9.57** | 8.59 | 8.20 | 36.85 | **9.21** |
> | gemma | 14.13 | 8.56 | 5.70 | 4.59 | 32.98 | 8.25 |
> | phi   | 18.10 | 7.12 | 6.00 | 5.07 | 36.29 | 9.07 |
>
> ##### Document Δ (per query)
>
> | model | step2_avg | step3_avg | step4_avg | step5_avg | total_sum | overall_mean |
> |:------|:---------:|:---------:|:---------:|:---------:|:---------:|:------------:|
> | 70b   | 1.93 | 1.44 | 1.15 | 1.25 | 5.77 | 1.44 |
> | 8b    | **2.12** | **1.65** | 1.53 | 1.32 | **6.62** | **1.66** |
> | gemma | 2.22 | 1.61 | 1.27 | 1.12 | 6.22 | 1.55 |
> | phi   | 2.03 | 0.69 | 1.04 | 0.95 | 4.71 | 1.18 |
>
>
> The first table shows the average number of newly generated keywords per query at each iteration (from step 2 to step 5), along with the total and mean values across all steps. The second table reports the same statistics for newly retrieved documents.
>
> From these results, we observe that each iteration contributes a non-trivial amount of new information. Specifically, each step generates on average:
>
> Keywords: approximately 3–18 new terms per iteration, resulting in a total of 20–37 new keywords per query across five steps, depending on the model.
>
> Documents: approximately 0.7–2.2 new documents per iteration, resulting in a total of 4.7–6.6 new documents per query over the full loop.
>
> These numbers confirm that later iterations continue to introduce new keywords and retrieve novel documents, supporting the utility of iterative refinement beyond the initial step.
>
> Thus, IterKey is flexibly designed to conserve computation for queries that do not require further processing, while enhancing accuracy and coverage through additional iterations when necessary. Therefore, the fact that the average number of iterations is relatively low should not be taken to imply that iteration is unnecessary, but rather that IterKey achieves a balance between efficiency and robustness through adaptive control.

---

> ### Author Response · Authors · 2025-06-01
> **The Response to the Comment from Reviewer NWKh 3/3**
>
> >High latency and consequently cost.
>
> >In Table 10, the runtime of components is extremely high? Is it total per dataset? I would have been more informative to understand latency per 1 example
>
> We acknowledge the concern about increased runtime, but believe the modest overhead is justified by the significant accuracy and robustness gains.
>
> As shown in Sec. 4.5, our framework caps iterations at 5, with most queries converging in under 1.5 iterations on average (Table 9), keeping latency low. IterKey is also more efficient than comparable baselines—for example, on EntityQA, it is about 400 seconds faster than ITRG while achieving higher accuracy (Table 10), demonstrating strong real-world efficiency and effectiveness.
>
> Finally, IterKey is model efficient: smaller models like LLaMA 3 8B perform comparably to LLaMA 3 70B across tasks (Table 2), making it scalable and cost effective even with limited resources.
>
> ---
>
> To quantify scalability, we followed the reviewer’s suggestion and computed the **per-question latency** by dividing total runtime by 500 samples, then multiplying by the average iteration count (1.5):
>
> | Step                  | RAG(BM25) | RAG(E5)  | ITRG     | IterKey  |
> |-----------------------|-----------|----------|----------|----------|
> | Query Expansion       | --        | --       | --       | 1.33 s   |
> | Retrieval             | 0.44 s    | 0.11 s   | 0.88 s   | 0.33 s   |
> | Answer Generation     | 0.64 s    | 0.63 s   | 3.39 s   | 1.05 s   |
> | Answer Validation     | --        | --       | --       | 0.72 s   |
> | **Total (per sample)**| **1.08 s**| **0.73 s**| **4.26 s**| **3.43 s**|
>
> - **3.43 s ✖ 1.5 = 5.15 s** per question on average for IterKey.
>
> Summary: While iterative reasoning introduces some overhead, IterKey’s end-to-end latency (~5.15 s per question with 1.5 iterations on average) remains well within acceptable bounds for real-time QA using GPU-based inference. Although this is about 4 s slower than non-iterative baselines, the substantial accuracy and robustness gains make the trade-off between runtime and answer quality justified in practical settings where reliability and interpretability are essential.
>
> >What is the explanation for IterKey(BM25) outperforming IterKey(E5) in Table 3, despite E5 being a stronger retriever in single-shot settings?
>
> Thank you for the question. This result stems from the keyword based retrieval formulation used in IterKey. In our approach, the query is created by concatenating the original question with an explicit list of keywords. This structure aligns well with sparse retrievers such as BM25, which score based on exact lexical matches and benefit directly from keyword presence and specificity.
>
> In contrast, dense retrievers such as E5 are trained on semantically rich natural language inputs and tend to struggle with keyword lists that lack syntactic and contextual flow. As a result, their performance decreases when used with keyword centric queries.
> By comparison, ITRG (Feng et al., 2024) generates full sentence query expansions using large language models, similar to Query2doc, which include more context and often candidate answers. These natural language reformulations are well suited for dense retrieval, allowing models such as E5 to perform effectively.
>
> As shown in Figure 2 in our paper, Row 3, ITRG using E5 achieves over 10 percent higher Top 1 to Top 3 recall than IterKey, highlighting the advantage of using dense retrievers with full sentence queries. This performance gap highlights a mismatch between keyword based queries and dense retrievers like E5.

---

> ### Author Response · Authors · 2025-06-01
> **Positioning IterKey Within Query Expansion Research**
>
> To improve clarity and prevent potential misunderstanding, we have revised the method description to explicitly position IterKey as an extension of prior work on query expansion.
> In the original version, the connection to existing QE methods such as Q2E and Query2doc was not clearly stated, which may have obscured the novelty and contribution of our approach.
>
> To address this, we now revise the paragraph to read as follows:
>
> **Before (Line 50 in Sec 2)**
>
> To address this limitation, we propose IterKey, a method that leverages LLMs to iteratively refine the retrieval process in RAG. Specifically, IterKey uses the self-validation capabilities of LLMs (Wang et al., 2024) to improve sparse retrieval through an iterative process of keyword generation, document retrieval, answer generation, and validation.
>
> **After**
>
> To address this limitation, we propose IterKey, a method that builds upon and extends query expansion research (e.g., Q2E; Jagerman et al., 2023; Query2doc; Wang et al., 2023) by leveraging LLMs to iteratively refine the retrieval process in RAG. Specifically, IterKey uses the self-validation capabilities of LLMs (Wang et al., 2024) to improve sparse retrieval through an iterative process of keyword generation, document retrieval, answer generation, and validation.
>
>
> [1] Jagerman et al., 2023. Query Expansion by Prompting Large Language Models.
>
> [2] Wang et al., 2023. Query2doc: Query Expansion with Large Language Models.

---

> ### Author Response · Authors · 2025-06-01
> **Discussion on Iteration Count and Retrieval Performance**
>
> To further strengthen our response **The Response to the Comment from Reviewer NWKh 2/3**, we include accuracy-by-iteration results in the table below.
>
> The following table reports the accuracy at each iteration step (in %):
>
> | Model     | Iter 1 | Iter 2 | Iter 3 | Iter 4 | Iter 5 |
> |-----------|--------|--------|--------|--------|--------|
> | LLaMA-8B  | 62.2   | 65.4   | 69.6   | 70.5   | 72.6   |
> | LLaMA-70B | 65.7   | 70.2   | 73.5   | 74.4   | 75.7   |
> | Gemma     | 53.2   | 57.6   | 60.0   | 61.2   | 63.2   |
> | Phi       | 57.5   | 64.5   | 67.9   | 68.8   | 69.6   |
>
> As shown, most performance gains are achieved in the first 1–2 iterations, where accuracy increases by up to 7 %. After the third iteration, improvements taper off, indicating diminishing returns.
> This confirms that IterKey balances efficiency and effectiveness by capturing most of the benefits early on, while retaining the flexibility to continue when necessary.

---

> > ### Comment · Reviewer_NWKh · 2025-06-05
> > **Response to authors**
> >
> > I thank the authors for their detailed responses. These responses broadly address the issues and questions I raised during the review. I encourage the authors to incorporate the suggested modifications and clarifications noted in the comments.

---

> > > ### Author Response · Authors · 2025-06-06
> > > **Official Comment by Authors**
> > >
> > > Thank you very much for your thoughtful and insightful comments.
> > > We sincerely appreciate your valuable suggestions and promise to incorporate them to improve the revised version of our paper.
> > > Thank you again for taking the time to review our work.

---

### Official Review · Reviewer_KgZB · 2025-05-11

**Rating:** 6
**Confidence:** 4
**Ethics Flag:** 1

**Summary:**

This paper introduces IterKey, an LLM-driven iterative keyword generation framework that enhances RAG via sparse retrieval. The primary goal of this paper is to balance the accuracy and interpretability by leveraging iterative keyword generation and self-evaluation during RAG process. While the paper is well-designed, there are certain significant concerns that need to be addressed.

**Questions To Authors:**

-	First of all, what is the difference between your proposed method and hybrid-based RAG (i.e., RAG that combines search results from sparse and dense vectors)? A clear explanation is needed to understand the novelty and improvement.
-	It seems that the retrieval efficiency highly depends on the number of iterations, but I could not find a discussion about this issue.
-	In step 1, the terms “explicit” and “implicit” are mentioned, but their meanings are not clearly defined. The explanation provided in the paragraph is vague, and the subsequent example does not sufficiently clarify the distinction. A more detailed explanation is needed.
-	In step 2, it is not clear how the enhanced query is formed, is it simply a concatenation of the original query and the generated keywords or what? Figure 1 should be revised to illustrate this step more clearly.
-	In step 3, “LLM verifies whether the generated answer is correct based on the retrieved documents, ensuring the RAG process has been executed correctly.”, how do authors ensure the reliability of the LLM’s verification process. Btw, the caption of Table 13 in Appendix C.1 mentions the keyword generation and answer validation, but the table content does not reflect the keyword generation, which is confusing. It is also unclear whether the LLM is judging the RAG output based on retrieved documents or a golden answer. 1) If LLM judges RAG output based on the retrieved documents, it is redundant as the RAG itself involves LLM generating a response based on retrieved contexts, and you use another LLM to evaluate the output, the validation process cannot be guaranteed (especially considering the largest LLM used for evaluation is 70B in 4-bit mode). 2) If LLM judges RAG output based on golden answer, the prompt in Table 13 does not present a placeholder for the golden answer, which needs clarification. In short, how exactly does your LLM judge the output is correct or not?
-	To better understand the proposed framework, it would be helpful to share the inputs and outputs of each step in this query example “What is the name of the spacecraft …”
-	Table 16 is not cited and its mentions in the table caption do not appear in the table content (e.g. underlined values, bold values.)
-	Table 17 is also not cited in the paper.
-	Appendix D explains that the number of keywords is not critical to performance, but why does the subsequent section devote so much space to presenting the keyword distribution for each model? This seems unnecessary unless there is a correlation between keyword distribution and retrieval performance/efficiency, which is not clearly discussed. A more focused analysis in this regard would be more impactful.

**Reasons To Accept:**

-	The motivation is clear.
-	Good presentation.

**Reasons To Reject:**

-	There are several significant concerns in the paper that go beyond revision and would require substantial clarification and restructuring to address effectively.
-	The objective of this work is to balance the accuracy and interpretability during RAG. In the experiments, the authors verify the proposed method can improve the RAG accuracy, but the interpretability aspect is not verified.
-	The experiment is not sufficient. Why not compare the proposed method with other RAG frameworks? For example, AdaptiveRAG [1], HippoRAG [2], Beam Retrieval [3], etc.

[1] Jeong, S., Baek, J., Cho, S., Hwang, S. J., & Park, J. C. (2024). Adaptive-RAG: Learning to Adapt Retrieval-Augmented Large Language Models through Question Complexity. In 2024 Conference of the North American Chapter of the Association for Computational Linguistics: Human Language Technologies (pp. 7036-7050). Association for Computational Linguistics.
[2] Gutiérrez, B. J., Shu, Y., Gu, Y., Yasunaga, M., & Su, Y. (2024, January). Hipporag: Neurobiologically inspired long-term memory for large language models. In The Thirty-eighth Annual Conference on Neural Information Processing Systems.
[3] Zhang, J., Zhang, H., Zhang, D., Yong, L., & Huang, S. (2024, June). End-to-End Beam Retrieval for Multi-Hop Question Answering. In Proceedings of the 2024 Conference of the North American Chapter of the Association for Computational Linguistics: Human Language Technologies (Volume 1: Long Papers) (pp. 1718-1731).

---

> ### Author Response · Authors · 2025-06-02
> **The Response to the Comment from Reviewer KgZB 1/6**
>
> We deeply appreciate your insightful and detailed comments. Your careful observations, including those on our motivation and broader structure, have been immensely valuable. Thank you for generously taking the time to provide feedback that greatly contributed to improving our paper.
>
> >The objective of this work is to balance the accuracy and interpretability during RAG. In the experiments, the authors verify the proposed method can improve the RAG accuracy, but the interpretability aspect is not verified.
>
> Thank you for highlighting the importance of verifying interpretability. In our work, we ensure full transparency of the RAG process by exposing every intermediate step:
>
> Keyword Generation: Readers can see exactly which keywords the model used to retrieve documents.
>
> Document Retrieval: Each retrieved document is explicitly shown, allowing inspection of the evidence base.
>
> Answer Generation: The final answer is traceable back to those keywords and documents.
>
> We use the term “interpretability” because all components, including keywords, retrieved documents, and the generated answer, are fully visible, allowing users to follow and audit the entire reasoning chain. In other words, every decision point is exposed by design.
>
> We also ground our reliance on sparse, keyword-based retrieval in prior work demonstrating its interpretability advantages. For example, Llordes et al. (2023)[1] show that sparse methods like BM25 offer clear, human-readable explanations for ranking decisions—an affordance that dense retrieval lacks. By adopting sparse retrieval, we inherit these transparency benefits and make it straightforward to understand and control the RAG pipeline.
>
>
> To illustrate this process more concretely, we present a representative example below.
>
> **Question:** Who is the author of *The Constant Nymph*?
> **Answer:** Margaret Kennedy
>
> | Iteration | Keywords                                                                                                         | Model Output                                                                                             | Validation Output | Correct? |
> |-----------|------------------------------------------------------------------------------------------------------------------|----------------------------------------------------------------------------------------------------------|-------------------|----------|
> | 1         | Maugham, Somerset, Maugham's, The Constant Nymph, author, Constant, Nymph, Maugham Somerset, W Somerset Maugham |  The Constant Nymph is a novel by Somerset Maugham | False             | False    |
> | 2         | Maugham, W Somerset Maugham, Constant, author, The Constant Nymph, Somerset Maugham, Somerset                   | The Constant Nymph is a novel by William Somerset Maugham | False             | False    |
> | 3         | Nymph, author, The Constant Nymph, William, Somerset, Somerset Maugham, Maugham's, Constant, Maugham, dramatist | The Constant Nymph is a novel by Margaret Kennedy.                                                       | True              | True     |
>
> In the third iteration, the retrieved documents included a long list of literary works and their authors, such as:
> Docs：
> >**Margaret Kennedy – The Constant Nymph ; Sinclair Lewis – Arrowsmith ; Walter Lippmann – The Phantom Public ; Anita Loos – Gentlemen Prefer Blondes ; W. Somerset Maugham – The Painted Veil ; Thomas Mofolo – Chaka ; Eugenio Montale – Ossi di seppia ; Liam O'Flaherty – The Informer ; Baroness Orczy ; The Miser of Maida Vale ; A Question of Temptation ; William Plomer – Turbott Wolfe ; Marcel Proust – Albertine disparue ; Jean-Joseph Rabearivelo – L'Aube rouge (The Red Dawn) ; Henry Handel Richardson (Et Florence Robertson) – The Way Home (second part of The Fortunes of Richard Mahony) ; Kate Roberts – O gors y bryniau (Welsh**
>
> From this example, it is evident that every step—keyword generation, document retrieval, and answer validation—is transparent and auditable, thereby ensuring interpretability by design.
>
> [1] Llordes et al., 2023. Explain Like I Am BM25: Interpreting a Dense Model’s Ranked List with a Sparse Approximation.

---

> ### Author Response · Authors · 2025-06-02
> **The Response to the Comment from Reviewer KgZB 2/6**
>
> >The experiment is not sufficient. Why not compare the proposed method with other RAG frameworks? For example, AdaptiveRAG [1], HippoRAG [2], Beam Retrieval [3], etc.
>
> Thank you for pointing out those additional RAG frameworks. While Adaptive-RAG, HippoRAG, and Beam Retrieval are indeed interesting and valuable contributions, they address different design goals and evaluation settings than ours, so a direct empirical comparison would not yield meaningful insights in the context of our framework.
> By contrast, IterKey is designed specifically as a multi‐iteration query‐expansion plus answer‐validation framework. Because our goal is to compare against methods that perform iterative keyword refinement, we chose ITRG (Iterative Retrieval‐Generation)[3] as the strongest publicly available baseline. ITRG already subsumes single‐pass QE approaches such as Q2E [1] and Query2doc [2] by concatenating LLM‐generated pseudo‐documents in multiple iterations. IterKey departs from ITRG in two key ways:
>
> - An explicit answer‐validation loop that stops as soon as a correct answer is detected, preventing unnecessary iterations.
> - Joint optimization of keyword generation and retrieval across multiple iterations, rather than a single‐pass expansion.
>
> As shown in Table 6, IterKey outperforms ITRG in both accuracy and runtime, demonstrating clear advantages over existing query‐expansion methods. To avoid confusion, we will explicitly cite Q2E and Query2doc in the revised manuscript and clarify that ITRG already generalizes them, ensuring that our comparison implicitly covers those earlier techniques. Because ITRG already represents the state of the art in iterative QE, we believe there is no additional insight to be gained from re‐evaluating Adaptive-RAG, HippoRAG, or Beam Retrieval in this context.
>
> Below is how we plan to update the “baselines” paragraph in the paper:
>
> **Before (Line 98)**
>
> As a representative iterative baseline, we adopted ITRG (Iterative Retrieval-Generation Synergy), which combines dense retrieval with iterative query refinement. Replicating its setup enables a direct comparison that highlights the strengths of IterKey, particularly in keyword refinement and retrieval accuracy.
>
> **After**
>
> As a representative iterative baseline, we adopt ITRG (Iterative Retrieval-Generation Synergy; Feng et al., 2024), the strongest publicly available method in this line of work.  ITRG repeatedly concatenates LLM-generated free-form text with the original query to refine retrieval quality.  In contrast, IterKey generates explicit keyword lists and introduces an answer-validation step that halts iteration once the answer is deemed correct.  Notably, ITRG already subsumes earlier single-pass query expansion methods such as Q2E (Bao et al., 2023) and Query2doc (Zheng et al., 2023), making it a suitable and more advanced baseline for comparison.
>
>
> [1] Jagerman et al., 2023. Query Expansion by Prompting Large Language Models.
>
> [2] Wang et al., 2023. Query2doc: Query Expansion with Large Language Models.
>
> [3] Feng et al., 2024. Retrieval-Generation Synergy Augmented Large Language Models.
>
> >First of all, what is the difference between your proposed method and hybrid-based RAG (i.e., RAG that combines search results from sparse and dense vectors)? A clear explanation is needed to understand the novelty and improvement.
>
> Hybrid RAG runs a sparse retriever (e.g., BM25) and a dense retriever (e.g., embeddings) in parallel, then merges or reranks their outputs. This can improve recall, but it does not change or expand the query itself. In other words, hybrid RAG simply selects documents that score well under both sparse and dense retrieval, without adjusting keywords to better capture the user’s intent.
>
> By contrast, IterKey builds on query expansion research (e.g., Q2E, Query2doc) and uses an LLM to iteratively expand and refine the query. Each iteration adds newly inferred keywords to the query and regenerates answers using the updated query. This process gradually incorporates terms that would otherwise be overlooked, enabling more accurate retrieval.
>
> **Before (Line 50)**
>
> To address this limitation, we propose IterKey, a method that leverages LLMs to iteratively refine the retrieval process in RAG. Specifically, IterKey uses the self-validation capabilities of LLMs (Wang et al., 2024) to improve sparse retrieval through an iterative process of keyword generation, document retrieval, answer generation, and validation.
>
> **After**
>
> To address this limitation, we propose IterKey, a method that builds upon and extends query expansion research (e.g., Q2E; Jagerman et al., 2023; Query2doc; Wang et al., 2023) by leveraging LLMs to iteratively refine the retrieval process in RAG. Specifically, IterKey uses the self-validation capabilities of LLMs (Wang et al., 2024) to improve sparse retrieval through an iterative process of keyword generation, document retrieval, answer generation, and validation.

---

> ### Author Response · Authors · 2025-06-02
> **The Response to the Comment from Reviewer KgZB 3/6**
>
> >It seems that the retrieval efficiency highly depends on the number of iterations, but I could not find a discussion about this issue.
>
> To address this concern, we provide the following accuracy-by-iteration results:
> The following table reports the accuracy at each iteration step (in %):
>
> | Model     | Iter 1 | Iter 2 | Iter 3 | Iter 4 | Iter 5 |
> |-----------|--------|--------|--------|--------|--------|
> | LLaMA-8B  | 62.2   | 65.4   | 69.6   | 70.5   | 72.6   |
> | LLaMA-70B | 65.7   | 70.2   | 73.5   | 74.4   | 75.7   |
> | Gemma     | 53.2   | 57.6   | 60.0   | 61.2   | 63.2   |
> | Phi       | 57.5   | 64.5   | 67.9   | 68.8   | 69.6   |
>
> As shown, most performance gains are achieved in the first 1–2 iterations, where accuracy increases by up to 7 %. After the third iteration, improvements taper off, indicating diminishing returns.
> This confirms that IterKey balances efficiency and effectiveness by capturing most of the benefits early on, while retaining the flexibility to continue when necessary.
>
>
>
> >In step 1, the terms “explicit” and “implicit” are mentioned, but their meanings are not clearly defined. The explanation provided in the paragraph is vague, and the subsequent example does not sufficiently clarify the distinction. A more detailed explanation is needed.
>
> Thank you for the feedback. You’re right that the current wording is ambiguous and hard to follow, so in the revised version we will remove those terms and replace the section with the following:
>
> **Before**
>
> >**Given a user query $q$, the LLM is prompted to generate a set of critical keywords $\mathcal{K}^0$ that are essential for retrieving relevant documents. By understanding the BM25-based retrieval algorithm and the query's intent, an LLM extracts critical keywords from the initial query, capturing both explicit and implicit essential terms. This approach uncovers nuanced relationships that sparse retrieval might overlook, resulting in more effective document retrieval. In our example, for the query “What is the name of the spacecraft that first landed humans on the Moon?”, LLM generates important keywords like Moon landing, Spacecraft, First humans.**
>
> **After**
> >**Given a user query $q$, we prompt the LLM to generate a set of critical keywords $\mathcal{K}_0$ that are useful for retrieving relevant documents. The LLM infers and produces search-important terms based on its understanding of the BM25 retrieval algorithm and the query’s intent, even when those terms are not fully expressed in the initial query. This approach fills in subtle relationships that sparse retrieval might miss, enabling more effective document retrieval. In our example, for the query “What is the name of the spacecraft that first landed humans on the Moon?”, LLM generates important keywords like Moon landing, Spacecraft, First humans.**
>
> >In step 2, it is not clear how the enhanced query is formed, is it simply a concatenation of the original query and the generated keywords or what? Figure 1 should be revised to illustrate this step more clearly.
>
> Indeed, Step 2 simply concatenates the original query with the generated keywords. We will clarify this in the text and update Figure 1 accordingly. Thank you.
>
>
> >Table 16 is not cited and its mentions in the table caption do not appear in the table content (e.g. underlined values, bold values.)
> Table 17 is also not cited in the paper.
>
> Thank you for pointing this out.In the revised version, we will make the following changes:
>
> - In the main text (around Line 821), add the sentence:
>
> >“The results are shown in Table 16 and Table 17.”
>
> - We will add “Underlined values ” and “Bold values ” to the table.
>
> With these changes, both Table 16 and Table 17 are properly referenced in the text, and their captions correctly explain the underlining and bolding used to highlight top results.

---

> ### Author Response · Authors · 2025-06-02
> **The Response to the Comment from Reviewer KgZB 4/6**
>
> >In step 3, “LLM verifies whether the generated answer is correct based on the retrieved documents, ensuring the RAG process has been executed correctly.”, how do authors ensure the reliability of the LLM’s verification process. Btw, the caption of Table 13 in Appendix C.1 mentions the keyword generation and answer validation, but the table content does not reflect the keyword generation, which is confusing. It is also unclear whether the LLM is judging the RAG output based on retrieved documents or a golden answer. 1) If LLM judges RAG output based on the retrieved documents, it is redundant as the RAG itself involves LLM generating a response based on retrieved contexts, and you use another LLM to evaluate the output, the validation process cannot be guaranteed (especially considering the largest LLM used for evaluation is 70B in 4-bit mode). 2) If LLM judges RAG output based on golden answer, the prompt in Table 13 does not present a placeholder for the golden answer, which needs clarification. In short, how exactly does your LLM judge the output is correct or not?
>
> Clarification Regarding Table 13
> Table 13 in Appendix C.1 covers only answer validation—not keyword generation. We apologize for the confusion. In this study, validation is based exclusively on retrieved documents, not a golden answer. We have updated the prompt to explicitly include the placeholder for “Retrieval Documents: {Docs}” so that it is clear the LLM judges correctness based on those documents alone.
>
> We would like to take this opportunity to clarify the rationale and practical value of this component in our framework.
>
> **1. Separation of Generation and Validation is Principled and Effective**
>
> Answer generation and answer validation serve fundamentally different purposes. Generation is inherently open-ended and often requires complex integration of information, whereas validation is a constrained binary task: determining whether the answer is logically supported by the retrieved documents. Even when using the same LLM, validation tends to be more reliable and interpretable, as demonstrated in frameworks such as LLM-as-a-Judge and Self-RAG [1]. Our framework adopts this principled separation of roles.
>
> **2. Validation Based Solely on Retrieved Documents is Sufficiently Robust**
>
> Originally, we validated using only the question and answer “Q and A”. Based on reviewer feedback and further experiments, we found that including retrieved documents “Q and A and Docs” provides a more principled and appropriate setup. We plan to make this the default in the revised version.
>
> While LLMs may lack domain or up to date knowledge, our validation step does not rely on the model’s internal knowledge. It only checks whether the answer is supported by the retrieved documents. This design ensures transparency and reproducibility. As long as retrieval is accurate, in context validation is sufficient and even preferable for interpretability and consistency. A human given the same documents would likely reach the same conclusion.
>
> To validate this design choice, we compared both prompt configurations across all models. As shown in Table 13, smaller models (e.g., Gemma-2, Phi-3.5) gain up to +2.3 pp when supporting documents are supplied, confirming that explicit evidence boosts validation accuracy. For larger models (LLaMA-3 8B/70B), the two settings deliver comparable accuracy, indicating that including documents neither degrades performance nor alters the overall conclusions.
>
> **3. Validation is Not Redundant, but an Essential Safety Check**
>
> In real-world applications, even when outputs are generated based on RAG, there remains a risk of misinformation if the retrieved documents are misleading or inappropriate. Therefore, a dedicated validation layer to verify whether the generated answer is truly grounded in the retrieved evidence is crucial for ensuring reliability, verifiability, and factuality in deployed systems.
>
> While our original implementation primarily used the “Q + A” setting, we revisited the design based on reviewer feedback and plan to adopt the “Q + A + Doc” configuration as the default in the revised version.
>
>
> [1] Asai et al., 2023. SELF-RAG: Learning to Retrieve, Generate, and Critique through Self-Reflection.

---

> ### Author Response · Authors · 2025-06-02
> **The Response to the Comment from Reviewer KgZB 5/6**
>
> >To better understand the proposed framework, it would be helpful to share the inputs and outputs of each step in this query example “What is the name of the spacecraft …”
>
> Below are the inputs and outputs for each step:
>
> **Step 1**
>
> **Input：**
>
> >**System: You are an assistant that generates keywords for information retrieval.
> User: Generate a list of important keywords related to the Query: {"What is the name of the spacecraft that first landed humans on the Moon?"}.
> Focus on keywords that are relevant and likely to appear in documents for BM25 search in the RAG framework.
> Output the keywords as: ["keyword1", "keyword2", ...]
> Separate each keyword with a comma and do not include any additional text.**
>
> **Output：**
>
> >**["Moon landing", "First humans", "Spacecraft", "Name", "Moon", "Space"]**
>
> **Step 2**
>
> **Input：**
>
> >**System: You are an assistant that generates answers based on retrieved documents.
> User: Here is a question that you need to answer:
>       Query: {"What is the name of the spacecraft that first landed humans on the Moon?"}
>       Below are some documents that may contain information relevant to the question.
>       Consider the information in these documents while combining it with your own knowledge to answer the question accurately.
>       Documents: {D}
>       Provide a clear and concise answer. Do not include any additional text.**
>
> **Output：**
>
> >**Space Shuttle Challenger**
>
>
> **Step 3**
>
> **Input：**
>
> >**System: You are an assistant that validates whether the provided answer is correct.
> User: Is the following answer correct?
>       Query: {"What is the name of the spacecraft that first landed humans on the Moon?"}
>       Answer: {Space Shuttle Challenger}
>       Retrieved Documents: {D}
>       Respond "True" or "False". Do not provide any additional explanation or text.**
>
> **Output：**
>
> >**False**
>
> **Step 4**
>
> **Input：**
>
> >**System: You refine keywords to improve document retrieval for BM25 search in the RAG framework.
> User: Refine the keyword selection process to improve the accuracy of retrieving documents with the correct answer.
>       Query: {"What is the name of the spacecraft that first landed humans on the Moon?}
>       Previous Keywords: {"Moon landing", "First humans", "Spacecraft", "Name", "Moon", "Space"}
>       Provide the refined list of keywords in this format: ["keyword1", "keyword2", ...].
>       Separate each keyword with a comma and do not include any additional text.**
>
> **Output：**
>
> >**["Moon landing", "Apollo 11", "Spacecraft", "Lunar module name", "Apollo program"]**
>
> **Step 2**
>
> **Input：**
>
> >**System: You are an assistant that generates answers based on retrieved documents.
> User: Here is a question that you need to answer:
>       Query: {"What is the name of the spacecraft that first landed humans on the Moon?"}
>       Below are some documents that may contain information relevant to the question.
>       Consider the information in these documents while combining it with your own knowledge to answer the question accurately.
>       Documents: {D}
>       Provide a clear and concise answer. Do not include any additional text.**
>
> **Output：**
>
> >**Eagle**

---

> ### Author Response · Authors · 2025-06-02
> **The Response to the Comment from Reviewer KgZB 6/6**
>
> >Appendix D explains that the number of keywords is not critical to performance, but why does the subsequent section devote so much space to presenting the keyword distribution for each model? This seems unnecessary unless there is a correlation between keyword distribution and retrieval performance/efficiency, which is not clearly discussed. A more focused analysis in this regard would be more impactful.
>
> Thank you for your observation. You are correct that Appendix D states that the number of keywords does not significantly affect performance, yet the following section dedicates considerable space to showing keyword distribution for each model. To address this inconsistency, we plan to replace that section with a focused analysis of how keyword and document counts change from one iteration to the next.
>
>
> ##### Keyword Δ (per query)
>
> | model | step2_avg | step3_avg | step4_avg | step5_avg | total_sum | overall_mean |
> |:------|:---------:|:---------:|:---------:|:---------:|:---------:|:------------:|
> | 70b   | 9.34 | 3.65 | 3.78 | 3.26 | 20.03 | 5.01 |
> | 8b    | **10.49** | **9.57** | 8.59 | 8.20 | 36.85 | **9.21** |
> | gemma | 14.13 | 8.56 | 5.70 | 4.59 | 32.98 | 8.25 |
> | phi   | 18.10 | 7.12 | 6.00 | 5.07 | 36.29 | 9.07 |
>
> ##### Document Δ (per query)
>
> | model | step2_avg | step3_avg | step4_avg | step5_avg | total_sum | overall_mean |
> |:------|:---------:|:---------:|:---------:|:---------:|:---------:|:------------:|
> | 70b   | 1.93 | 1.44 | 1.15 | 1.25 | 5.77 | 1.44 |
> | 8b    | **2.12** | **1.65** | 1.53 | 1.32 | **6.62** | **1.66** |
> | gemma | 2.22 | 1.61 | 1.27 | 1.12 | 6.22 | 1.55 |
> | phi   | 2.03 | 0.69 | 1.04 | 0.95 | 4.71 | 1.18 |
>
>
> The first table shows the average number of newly generated keywords per query at each iteration (from step 2 to step 5), along with the total and mean values across all steps. The second table reports the same statistics for newly retrieved documents.
>
> From these results, we observe that each iteration contributes a non-trivial amount of new information. Specifically, each step generates on average:
>
> Keywords: approximately 3–18 new terms per iteration, resulting in a total of 20–37 new keywords per query across five steps, depending on the model.
>
> Documents: approximately 0.7–2.2 new documents per iteration, resulting in a total of 4.7–6.6 new documents per query over the full loop.
>
> These numbers confirm that later iterations continue to introduce new keywords and retrieve novel documents, supporting the utility of iterative refinement beyond the initial step.
>
> Thus, IterKey is flexibly designed to conserve computation for queries that do not require further processing, while enhancing accuracy and coverage through additional iterations when necessary. Therefore, the fact that the average number of iterations is relatively low should not be taken to imply that iteration is unnecessary, but rather that IterKey achieves a balance between efficiency and robustness through adaptive control.

---

> ### Author Response · Authors · 2025-06-06
> **Kind reminder**
>
> Dear reviewer,
>
> We have sincerely addressed your concerns in our rebuttal. As we haven’t heard back during the discussion, we would be grateful if you could kindly take a moment to review our replies.
>
> If our responses have resolved your concerns, we would appreciate it if you could reflect that in your final score. Of course, we are happy to provide further clarification if needed.
>
> Thank you again for your time and thoughtful feedback.
>
> Authors

---

> > ### Comment · Reviewer_KgZB · 2025-06-07
> >
> > Thank you for the detailed responses to each point. I have increased my score. Hope that these changes and new results can be added in the next version.

---

> > > ### Author Response · Authors · 2025-06-07
> > > **Official Comment by Authors**
> > >
> > > Thank you very much for your kind follow-up and for raising your score.
> > >
> > > Your insightful and objective feedback greatly improved our paper. We sincerely appreciate the time and care you devoted to it.
> > >
> > > Thank you again for your thoughtful and constructive comments.

---

### Official Review · Reviewer_cbBx · 2025-05-11

**Rating:** 6
**Confidence:** 4
**Ethics Flag:** 1

**Summary:**

The paper introduces IterKey, an LLM-driven iterative keyword generation framework designed to enhance Retrieval-Augmented Generation (RAG) by combining sparse retrieval (BM25) with iterative refinement. IterKey operates through three stages: keyword generation, answer generation, and answer validation, iterating until a validated answer is obtained. Experiments on four QA datasets (NQ, EntityQA, WebQA, HotpotQA) demonstrate that IterKey improves BM25-based RAG by up to 20% in accuracy, achieving performance comparable to dense retrieval methods (e.g., E5) and prior iterative approaches (e.g., ITRG). The method leverages LLMs to balance accuracy and interpretability while preserving the transparency of sparse retrieval.

**Reasons To Accept:**

- Strong Empirical Results: Demonstrates significant improvements over BM25 baselines (5–20% accuracy gains) and competitive performance against dense retrieval methods across multiple models (Llama-3.1, Gemma-2, Phi-3.5).

**Reasons To Reject:**

- Model Dependency: Performance heavily relies on the LLM’s capabilities (e.g., Gemma-2 underperforms despite format compliance), limiting generalizability to weaker or smaller models.

- Computational Overhead: Iterative process increases runtime (1.5–5x slower than non-iterative RAG), raising concerns for scalability and cost in production settings.

- Limited Theoretical Insight: Empirical observations (e.g., optimal keyword counts, layer sensitivity) lack theoretical grounding, leaving open questions about why certain design choices work.

---

> ### Author Response · Authors · 2025-06-01
> **The Response to the Comment from Reviewer cbBx 1/2**
>
> Thank you very much for your thoughtful and detailed feedback. We appreciate the time and effort you have taken to help us improve our work.
>
> >Strong Empirical Results: Demonstrates significant improvements over BM25 baselines (5–20% accuracy gains) and competitive performance against dense retrieval methods across multiple models (Llama-3.1, Gemma-2, Phi-3.5).
>
>
> We thank the reviewer for highlighting the concern regarding model dependency. Our response is as follows:
>
> In practical deployments, the choice of LLM is often predetermined based on application constraints (e.g., available compute, latency requirements). Our framework, therefore, is designed under the assumption that a specific model is selected during system design. In this context, model choice is not an external limitation but rather an integral component of real-world RAG systems.
>
> While it is true that model capabilities affect performance, our analysis reveals that the primary bottleneck lies in the instruction-following and validation abilities of LLMs, particularly in the answer verification step. As demonstrated in Tables 4 and 5, when the validation step is conducted by a more capable model (e.g., LLaMA-3-70B), even underperforming models like Gemma-2 can achieve significant accuracy gains  +8.3%. This underscores the importance of validation quality, not merely model size.
>
> Moreover, we observe that smaller models such as LLaMA-3-8B can achieve performance levels comparable to LLaMA-3-70B in multiple tasks (Table 2). This demonstrates that IterKey is not inherently dependent on model scale, but instead benefits from models with stronger reasoning and verification capabilities.
>
> In summary, we argue that our method’s contribution lies not in masking model weaknesses, but in exposing which models are suitable for instruction-following tasks such as IterKey. This provides actionable insights for model selection in downstream deployments.
>
>
> >Computational Overhead: Iterative process increases runtime (1.5–5x slower than non-iterative RAG), raising concerns for scalability and cost in production settings.
>
>
> We acknowledge the concern about increased runtime, but believe the modest overhead is justified by the significant accuracy and robustness gains.
>
> As shown in Sec. 4.5, our framework caps iterations at 5, with most queries converging in under 1.5 iterations on average (Table 9), keeping latency low. IterKey is also more efficient than comparable baselines—for example, on EntityQA, it is about 400 seconds faster than ITRG while achieving higher accuracy (Table 10), demonstrating strong real-world efficiency and effectiveness.
>
> Finally, IterKey is model efficient: smaller models like LLaMA 3 8B perform comparably to LLaMA 3 70B across tasks (Table 2), making it scalable and cost effective even with limited resources.
>
> ---
>
> To quantify scalability, we followed the reviewer’s suggestion and computed the per-question latency by dividing total runtime by 500 samples, then multiplying by the average iteration count (1.5):
>
> | Step                  | RAG(BM25) | RAG(E5)  | ITRG     | IterKey  |
> |-----------------------|-----------|----------|----------|----------|
> | Query Expansion       | --        | --       | --       | 1.33 s   |
> | Retrieval             | 0.44 s    | 0.11 s   | 0.88 s   | 0.33 s   |
> | Answer Generation     | 0.64 s    | 0.63 s   | 3.39 s   | 1.05 s   |
> | Answer Validation     | --        | --       | --       | 0.72 s   |
> | **Total (per sample)**| **1.08 s**| **0.73 s**| **4.26 s**| **3.43 s**|
>
> - **3.43 s ✖ 1.5 = 5.15 s** per question on average for IterKey.
>
> Summary: While iterative reasoning introduces some overhead, IterKey’s end-to-end latency (~5.15 s per question with 1.5 iterations on average) remains well within acceptable bounds for real-time QA using GPU-based inference. Although this is about 4 s slower than non-iterative baselines, the substantial accuracy and robustness gains make the trade-off between runtime and answer quality justified in practical settings where reliability and interpretability are essential.

---

> > ### Comment · Reviewer_cbBx · 2025-06-06
> >
> > Thanks for authors' rebuttal content. After carefully reviewing the answers of weaknesses, I increase my score.

---

> > > ### Author Response · Authors · 2025-06-06
> > > **Official Comment by Authors**
> > >
> > > We truly appreciate your thoughtful feedback and the time you dedicated to reviewing our work.
> > > Thank you again for your kind and constructive engagement.

---

> ### Author Response · Authors · 2025-06-01
> **The Response to the Comment from Reviewer cbBx 2/2**
>
> >Limited Theoretical Insight: Empirical observations (e.g., optimal keyword counts, layer sensitivity) lack theoretical grounding, leaving open questions about why certain design choices work.
>
> While our study is primarily empirical, each major component of the IterKey framework is grounded in prior research. Iterative keyword generation is supported by recent work on LLM-based query expansion and pseudo-document generation, which show that LLMs can effectively identify semantically rich terms to enhance retrieval recall [1][2]. Similarly, our validation step is aligned with frameworks such as Self-RAG and LLM-as-a-Judge, which demonstrate the effectiveness of separating answer generation and verification for improved factuality [3][5].
>
> The decomposition of reasoning into separate stages such as keyword generation, answer generation, and validation is inspired by insights from Chain of Thought prompting [4], which improves structured and step-by-step reasoning, and from ReAct style prompting [5], which allows reasoning and acting to be interleaved for better factual grounding. In addition, our use of sparse keyword retrieval is supported by recent studies that emphasize its advantages in interpretability and controllability compared to dense methods in real world RAG applications [6].
>
> Therefore, although we do not present formal theoretical models, our design choices are **principled and informed by a rich body of existing literature**. We view our framework as a synthesis of these insights, and we believe it provides a valuable foundation for future theoretical exploration of iterative, interpretable RAG pipelines, where each stage is guided by well-established design intuitions.
>
> [1] Xia et al., 2025. Knowledge-Aware Query Expansion with Large Language Models for Textual and Relational Retrieval.
>
> [2] Mackie et al., 2023. Generative Relevance Feedback with Large Language Models.
>
> [3] Asai et al., 2023. SELF-RAG: Learning to Retrieve, Generate, and Critique through Self-Reflection.
>
> [4] Wei et al., 2022. Chain-of-Thought Prompting Elicits Reasoning in Large Language Models.
>
> [5] Yao et al., 2023. ReAct: Synergizing Reasoning and Acting in Language Models.
>
> [6] Llordes et al., 2023. Explain Like I Am BM25: Interpreting a Dense Model’s Ranked List with a Sparse Approximation.

---

> ### Author Response · Authors · 2025-06-06
> **Kind reminder**
>
> Dear reviewer,
>
> We have sincerely addressed your concerns in our rebuttal. As we haven’t heard back during the discussion, we would be grateful if you could kindly take a moment to review our replies.
>
> If our responses have resolved your concerns, we would appreciate it if you could reflect that in your final score. Of course, we are happy to provide further clarification if needed.
>
> Thank you again for your time and thoughtful feedback.
>
> Authors

---

### Comment · Program_Chairs · 2025-04-03

This paper violates the page limit due to adding a limitation sections beyond the page limit. COLM does not have a special provision to allow for an additional page for the limitations section. However, due to this misunderstanding being widespread, the PCs decided to show leniency this year only. Reviewers and ACs are asked to ignore any limitation section content that is beyond the 9 page limit. Authors cannot refer reviewers to this content during the discussion period, and they are not to expect this content to be read.

---

### Decision · Program_Chairs · 2025-07-08

**Decision:**

Accept

**Comment:**

The paper proposes a RAG approach that uses iterative keyword generation for improved interpretability of the RAG pipeline. The reviewers appreciated the idea, but had multiple concerns in the initial review cycle. These were largely addressed in the author rebuttal and reviewers are now in support of accepting.